# Landscape Is Alive: Nuwuvi Pilgrimage and Power Places in Nevada

**Richard Stoffle** [1,*], **Richard Arnold** [2] **and Kathleen Van Vlack** [3]

1    School of Anthropology, University of Arizona, Tucson, AZ 85718, USA
2    Pahrump Paiute Tribe, Pahrump, NV 89041, USA; rwarnold@hotmail.com
3    Living Heritage Research Council, Cortez, CO 81321, USA; kvanvlac@email.arizona.edu
*    Correspondence: rstoffle@arizona.edu; Tel.: +1-520-907-2330

**Abstract:** Cultural landscapes are defined at Creation, according to the beliefs of the *Nuwuvi* (Paiute) and Newe (Shoshone peoples). After Creation, the Native people came to understand the purpose of *living* landscapes and special places within them. During this time, some places that were designated as essential parts of landscapes at Creation had been inscribed by Native people with peckings and paintings and honored with offerings. Special spiritual places within the landscape were networked like the pearls on a string to produce the foundation of pilgrimage trails. This is an analysis of one such valley landscape in southern Nevada, USA and a pilgrimage trail extending between the Pahranagat Valley and the Corn Creek oasis at the foot of the Paiute Origin place called *Nuvagantu* (Spring Mountains). Tribal representatives from 18 consulting tribes participated in a special environmental impact assessment to explain this landscape, its components, and potential impacts that could derive from it being removed from a wildlife refuge to become a part of a military land and air use area.

**Keywords:** living cultural landscapes; great basin pilgrimage trails; Southern Paiute; Western Shoshone; cultural heritage; environmental impact assessment



## 1. Introduction

This is an analysis based on Native American expert testimony regarding the existence and cultural centrality of a cultural landscape in Nevada that extends between Pahranagat Valley and Black Butte in the north to Corn Creek and the Spring Mountains in the south. Centered in the valley is a pilgrimage trail that passes from place to place. The analysis concludes that special places are inherently meaningful but can only be fully understood by the role they play in the formation and operation of *living* cultural landscapes and pilgrimage trails. In other words, cultural landscapes are the more robust interpretative frame for explaining both places and the spaces between them and should be the analytical center of land management.

The analysis is guided by the following epistemological postulates that were shared by tribal representatives during this study:

<div align="center">The Landscape Is Alive</div>

*Since the beginning of time, the Creator made Nuwuvi peoples at Nuvagantu and here became attached to this place and the place to us. The land is alive, which means that there is power in all things such as rocks, water, air, plants, animals, and humans. All of these beings are interconnected; they can talk with each other and work together*
*to balance the world.*

*The land has eyes and ears. It can talk and knows our thoughts. This makes it in balance and provides guidance. When treated badly, the mountains and everything within them suffers. Misuse of these areas upsets the balance and can cause great harm as well as diminish their power.*

The Mountains Are Part of Us

*It's not just a mountain to us, it's a living thing, living spirit to us, the trees and the rocks and the air, the water, they're all our cousins, part of us, related to us. So we're, it's just not a big mountain there, it's part of us. We're related.*

*It's alive.*

These cultural stipulations derive from the Native beliefs that, at Creation, all the elements of the Earth were given a life force, generally called *Puha.* This energy permits Earth elements to talk, hear, have desires, travel, and know their Creation-based responsibility to each other. By following these responsibilities, they maintain balance for themselves and the Earth. Everything has a purpose. For these reasons, these elements are seen as people in various relationships and treated with respect by *Nuwuvi* people. The elements tend to concentrate in special places and be functionally interconnected, like pearls on a string, across space and time and having a common purpose.

## 2. Background

This analysis contributes to the original Western academic definition of landscape [1,2] This definition focuses on the relationship between the formulation of a cultural landscape and the places within it, as described by the geographer Otto Schlüter. In 1908, Schlüter defined two forms of landscape: the *Urlandschaft* (original landscape) or a natural landscape that existed before major human induced changes and the *Kulturlandschaft* (cultural landscape), a landscape created by human culture [3]. Schlüter argued that the major task of geography was to trace the changes in these two landscapes [4].

In 1925, Carl Sauer wrote that cultural landscapes are made up of human forms and actions superimposed on physical landscapes. He was extremely influential in promoting and developing the idea that culture serves as a force in shaping the visible features of the Earth's surface in delimited areas. Within his definition, the physical environment retains a central significance, as the medium with and through which human cultures act [5]. He maintained that objects which exist together in the landscape exist in interrelation. His classic definition of a 'cultural landscape' reads as follows: "The cultural landscape is fashioned from a natural landscape by a cultural group. Culture is the agent, the natural area is the medium, the cultural landscape is the result".

Since Schlüter's first formal use of the term, and Sauer's effective promotion of the idea, the concept of *cultural landscapes* has been variously used, applied, debated, developed, and refined within academia. During the ensuing 100 years, the concept of natural or physical landscape continued to be debated, refined and incorporated into national heritage preservation [6]. The physical reality of landscape was emphasized by geographers who specialized in mapping [6]. The development of the concept of cultural landscape was more controversial because it often involved intangible components and was thus difficult or impossible to map. Early cultural landscape discussions focused on real human features such as the organization of villages in a countryside and historic gardens. Discussion with traditional, native, and aboriginal people gave rise to the concepts of evolved and culturally associated landscapes [6]. Research on both landscape categories involved researching local people or people who had lived in the landscape under study.

In 1992, the World Heritage Committee convened a meeting of landscape specialists to advise and assist redrafting the Committee's Operational Guidelines to include 'cultural landscapes' as an option for heritage listing properties that were neither purely natural nor purely cultural in form (i.e., mixed' heritage) [7]. At this meeting according to Smith [6]:

> The French and English requests for designating landscapes as World Heritage sites exposed weaknesses in the separation of natural and cultural heritage in the designation process. Because of the legal and financial implications of UNESCO designation, terminology became an important issue. The ICOMOS Landscapes Working Group, at its critical meeting in 1992 in France, adopted the term cultural landscape or *paysage culturel* to describe those "combined works of nature and of man" that are illustrative of the evolution of human society and settlement over time.

Building on international heritage preservation pressures for further conceptual clarification, according to Smith [6] in 2009, the World Heritage Committee developed three sub-categories of cultural landscape [8]:

- Designed cultural landscapes, often gardens or parks that are the work of a single designer or period, and that have clear aesthetic intent;
- Evolved cultural landscapes, which are the result of a gradual adaptation of a community to an environment, often through many generations, and that may be continuing (if still active) or relict (if no longer inhabited or active);
- Associative cultural landscapes, which have powerful religious, artistic or cultural associations and which may have little material culture evidence.

As local, native, and aboriginal people became directly involved in landscape studies, they insisted on the landscape studies and heritage denotation to reflect an overlap of natural and cultural heritage. In Canada and elsewhere, the term *aboriginal cultural landscape* came into use [6]. The concept of cultural landscapes, according to Smith [6], is not inherently physical in the observable sense—they exist in the cultural imagination. They are physical to the extent that they are experienced and that this experience becomes culturally shared. They are intangible as well as tangible, kept alive by a continual process of re-imagination and cultural practice. A cultural landscape that is healthy is not necessarily static, but rather in a state of equilibrium—a place with a healthy ecology. A healthy ecology is not so much about the visual appearance of a place but rather about its internal sense of balance. This analysis contributes to the on-going discussion of *living cultural landscapes*.

Native American people tend to view cultural resources as being bound together in broad categories based on functional interdependency and proximity rather than being solely defined by inherent characteristics [9]. Most places where Indian people traditionally lived and visited contained the diverse necessities of life such as plants and animals for food, medicinal plants for continued health, and paintings and peckings on rock walls that talk about historic events and broadly define the place's purpose. There are ceremonial blessings in the area where the people gathered, often near water used for drinking, for agriculture and horticulture, and for ceremonies of all kinds. Indian people perceive places and the things associated with them as Earth components that are organized and interrelated into what are called *cultural landscapes* [10,11]. Cultural landscape components with functionally different relationships may include: (1) archaeological sites that are both ceremonial and used for gathering plants and minerals, (2) animals that were used as food, but they also appear in rock paintings and peckings that depict the spiritual relationship between Indian people and animals, and (3) viewscapes of living mountains and playas that convey the timeless communication of Native people with the land.

Greider and Garkovitch [12] conclude that Native cultural groups socially construct landscapes as reflections of themselves. In this process, the social, cultural, and natural environments are meshed and become part of the shared symbols and beliefs of members of the groups. Thus, the natural environment changes because it takes on different meanings depending on the social and cultural symbols affiliated with it. Many Native Americans, however, stipulate that the landscapes existed at Creation with their own purpose and humans have over time learned about these and incorporated them into their own purposes. From the Native perspective, the landscapes have their own existence apart from humans and thus are not created by human social constructions. Indian people believe that they are largely reflections of their landscapes, which were created to guide and support them and bring balance to the Earth. People and landscapes thus formed and sustain each other.

The findings of this analysis are consistent with those from other Native American natural land and cultural place studies such as Deward Walker [13,14], Fikret Berkes [15], and Winona La Duke [16]. Our analysis closely parallels that of the recent study of Zuni Pueblo concerns for a proposed uranium mine [17]. Culturally affiliated Native American people stipulate that "It is not so much a particular place that matters but rather how that place fits within the landscape, how it connects to other important places, and what vistas can be taken in from within a place or when viewing it from a distance," [18]. Whether

using Walker's term Sacred Geography, Holy Lands [19], or just landscape, clearly common cultural connections to the land exist for most Native American people [20,21].

Our research has documented five major types of cultural landscapes as these are perceived by many American Indian peoples [9,11]. Categorized in terms of size and function, these types of Native American cultural landscapes and places are: (1) holy landscapes, (2) storyscapes, (3) regional landscapes, (4) ecoscapes, and (5) landmarks. Places tend to be landmarks in scale and function. Studies of these landscapes require different methodologies [22]. In "Landmark and Landscape: A Contextual Approach to the Management of American Indian Resources," Zedeño, Austin and Stoffle [23] concluded, based on multiple studies, that cultural landscapes are more readily understood if many interviews are conducted at various locations over several seasons and years.

Landscapes cannot be discussed in the absence of place knowledge, but such knowledge cannot be obtained with a few interviews using the same questions needed to define places. Places are also networked through various connections, which create synergistic relationships that increase the complexity and difficulty of understanding both themselves and their roles in cultural landscapes [23].

The act of pilgrimage involves a person or a group of people traveling along trail from place to place usually for a great distance. Travelers stop at shrines and prayer places to prepare for their ultimate ceremonial destination where they seek spiritual enlightenment and participate in balancing ceremonies [24,25]. Pilgrimage participants formed relationships with each other, the places they visit, and the objects used in ritual during their journey [26]. Numic speaking people have long asserted that these relationships need to be considered to understand pilgrimage and the places that comprise its cultural landscape [27,28].

It is further understood that Native places and landscapes receive non-Native social constructions especially when multiple cultural groups view them as culturally significant and proceed to provide their own interpretations [29]. Cultural landscapes are often renamed such as *Puhant 'uvipe*, which means *Big River Canyon* in Paiute but is now called the Grand Canyon [30], and *Ha'tata*, which means *The Backbone of the River* in the Hualapai language and is now called Hoover Dam and Lake Mead [31]. Both cases involve canyons with complex landscapes located along the Colorado River that are sacred to many Native peoples, but each has been renamed and reconceptualized as locations for dams to make reservoirs for electrical production and rivers to support rafting tourists. Non-Natives often reject Native definitions of places and landscapes. The cultural meaning of *Ha'tata*, for example, became a multiple agency debate wherein the Native perceptions were argued as not true and, thus, they could not influence the location of a bridge planned to span the river at Hoover Dam [32].

Like landscapes, when Native people strive to discuss the meaning of places, often there are alternative and conflicting social constructions of meanings and names, both in terms of the where places are (that is their boundaries in space, time, and dimensions), their cultural importance, and even whether they exist [33]. Devils Tower in Wyoming, for example, is a place known by many Native people to be a sacred portal to another dimension. It is called *Mato Tipila* which means Bear's Lodge in Lakota and *Na Kovehe* (Bear Lodge) in Cheyenne, but to non-Natives it is just a formidable and inert volcanic rock that is best used for climbing [34]. One such *place problem* in this study is the *playa* (i.e., an ancient dry lake), which is culturally significant as a place of aboriginal occupation and a path to other dimensions for Native peoples, but for others, the playa is a wasteland best used as a sacrifice zone.

## 3. The Study

The present analysis presents a cultural perspective that emerged from ethnographic interviews and Native American self-reporting. Here, we describe a traditional Native American cultural landscape that is centered on a pilgrimage trail. This is an ethnographic study of potential impacts from a proposed Land Withdrawal from a major U.S. wildlife refuge for use instead by one of the largest U.S. military facilities in the world. This

analysis is not about the cultural impacts of the potential Land Withdrawal, which are published as part of the final environmental assessment [35], but instead is focused on an ethnographic outcome of the study, which is the identification of a distinct valley landscape and pilgrimage trail with culturally special places and natural resources. This analysis documents why it is essential to identify from a Native American cultural perspective the broader integrating concepts of landscape and place to understand more specific cultural concerns about cultural elements such as animals, plants, archaeology, and topography.

The ethnographic study occurs on a portion of the aboriginal lands of the Southern Paiutes, Western Shoshone, and Owens Valley Paiutes and Shoshone peoples. These cultural groups are organized as 18 tribes and are officially in government-to-government consultation with the Nellis Air Force Base (Figure 1).

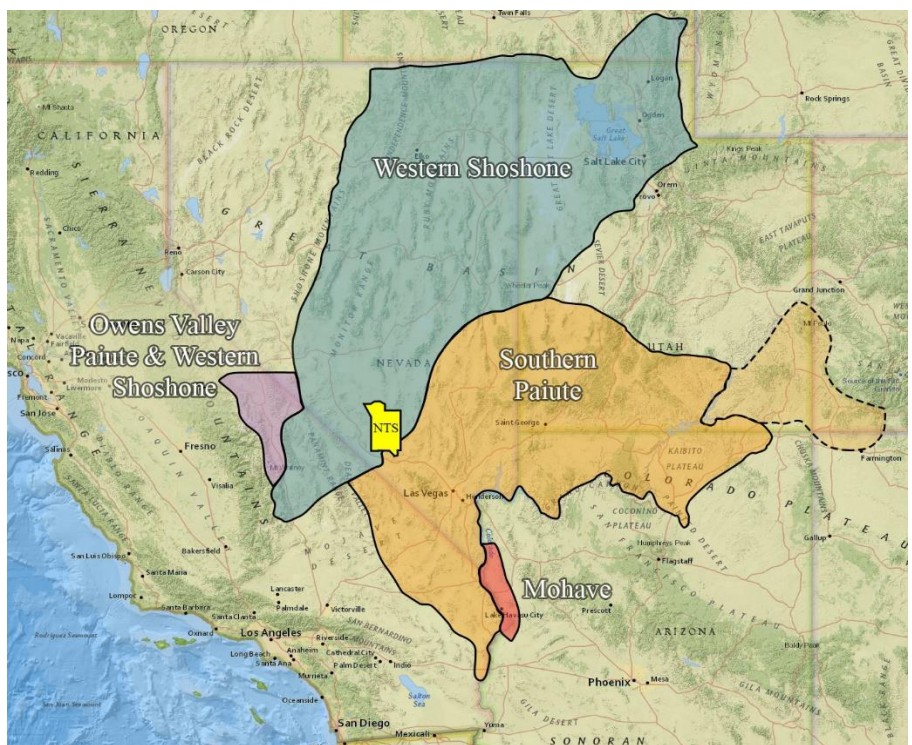

**Figure 1.** Map of Aboriginal Lands of Nellis Consulting Group of Tribes and Organizations.

The Nellis Air Force Base, Nevada Test and Training Range (NTTR) is the largest contiguous air and ground space available for peacetime military operations in the free world. The range occupies 2.9 million acres of land, 5000 square miles of airspace which is restricted from civilian air traffic over-flight, and another 7000 square miles of Military Operating Area, or MOA, which is shared with civilian aircraft [36].

This analysis derives from Native American ethnographic interviews and Native American self-reporting from a field study conducted in 2017 and 2018 [37]. The ethnographic study is an assessment of the impacts on Native resources, places, and landscapes of the potential expansion of NTTR to incorporate a large portion of the Desert National Wildlife Refuge [38] into the NTTR for exclusive military use [35]. As posted on this website, the proposed land withdrawal was to be evaluated by the U.S. Congress as a Legislative Environmental Impact Statement (LEIS):

> The LEIS is the detailed environmental statement required by law that will support the legislative proposal. The Air Force is the lead agency for the LEIS, while the BLM, the Department of Energy (DOE), which includes the Nevada National Security Site and the National Nuclear Security Administration, Nevada Department of Wildlife, and the U.S. Fish and Wildlife Service (USFWS) are cooperating agencies. The Air Force recognizes that there may be impacts to other

stakeholders and will have dialogue with the appropriate Nevada state agencies as well as local counties and cities that may be impacted by the land withdrawal. *The Air Force will also conduct government-to-government consultation with each of the federally recognized tribes potentially affected by the NTTR land withdrawal* (emphasis added).

The final italicized stipulation is the basis for funding this ethnographic study, which meets the federal requirement of government-to-government consultation [39].

The proposed land transfer area that is the focus of this ethnographic analysis was designated Alternative 3C (Figure 2), which is one of three potential land expansions by the U.S. Air Force for the NTTR. The proposed expansions are to be evaluated by the U.S. Congress as a special kind of environmental impact assessment called a Legislative Environmental Impact Statement (that is a LEIS). If approved by the U.S. Congress, most of the Desert National Wildlife Refuge (DNWR) would be transferred from its current management by the U.S. Fish and Wildlife Service and become a military air and land use area for the NTTR [35].

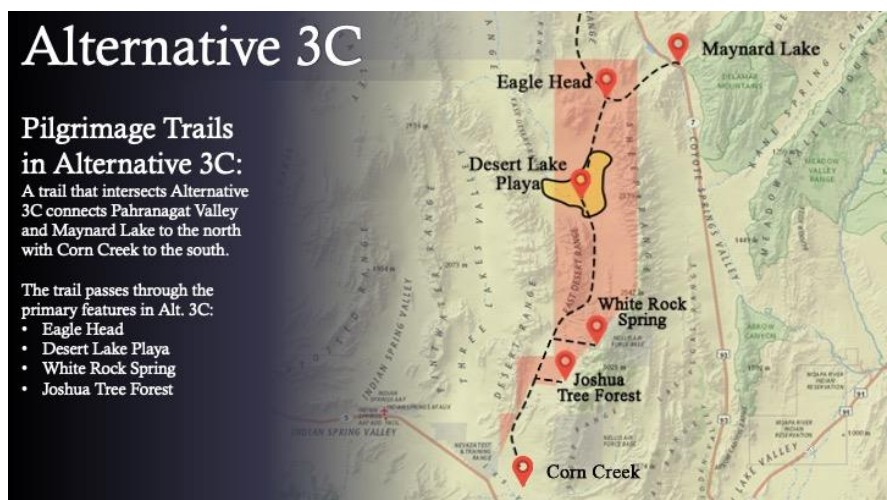

**Figure 2.** Alternative 3C in pink and places studied located along the pilgrimage trail [40].

The Nevada study area extends west to east from the Desert Range along the current eastern boundary of the UTTR. It also extends from the north to south trending along the crest of Sheep Mountains (Figure 3). It extends from north to south from Pahranagat Valley to the Spring Mountains. The pilgrimage trail extends from Pahranagat Valley and Black Butte to Eagle Head, to the Desert Lake Playa, along the trail to White Rock Spring, through the Joshua Tree Forest and is anchored in the south by Corn Creek and the Spring Mountains. Pilgrimage trails and secular travel trails have been identified throughout the broader area. Some of these trails are identified in the 2018 Nellis Integrated Report Meeting map below along with the boundary of the UTTR in tan and the DNWR in green.

Both secular and sacred traditional Native trails, including the Black Butte trail analyzed here, were identified during ethnographic mapping studies of the Spring Mountains [41] and the Pintwater Range and Cave [42]. Data on the relationships between places and trails were conducted with survey forms and large printed maps. Field testing, which was needed to more fully understand the trails and places along them, was not possible at the time. Subsequent studies, however, including this LEIS study, have provided these ethnographic details [37].

A key background factor that influenced the LEIS ethnographic interviews involved two massacres of Native people that occurred in the Pahranagat Valley in the mid-1860s. These events are remembered and have been discussed by tribal representatives in previous studies [43–45]. Some tribal representatives decided not to fully interpret the cultural landscape, selected places within it, or the pilgrimage trail by responding with short

answers. Most representatives, however, were more forthright about the meaning of the cultural landscape and the historic events that occurred there. During the study, some representatives expressed the opinion that the massacres and encroachment be remembered, and the places of genocide be memorialized. This is a process called the *Heritagization of Sites of* Suffering [46] and the visitation of such sites is called Dark Tourism. Such places of horror and shame are also called *Sites of Conscience* [47] and there is an international collation devoted to assuring their protection so they can serve to educate new generations.

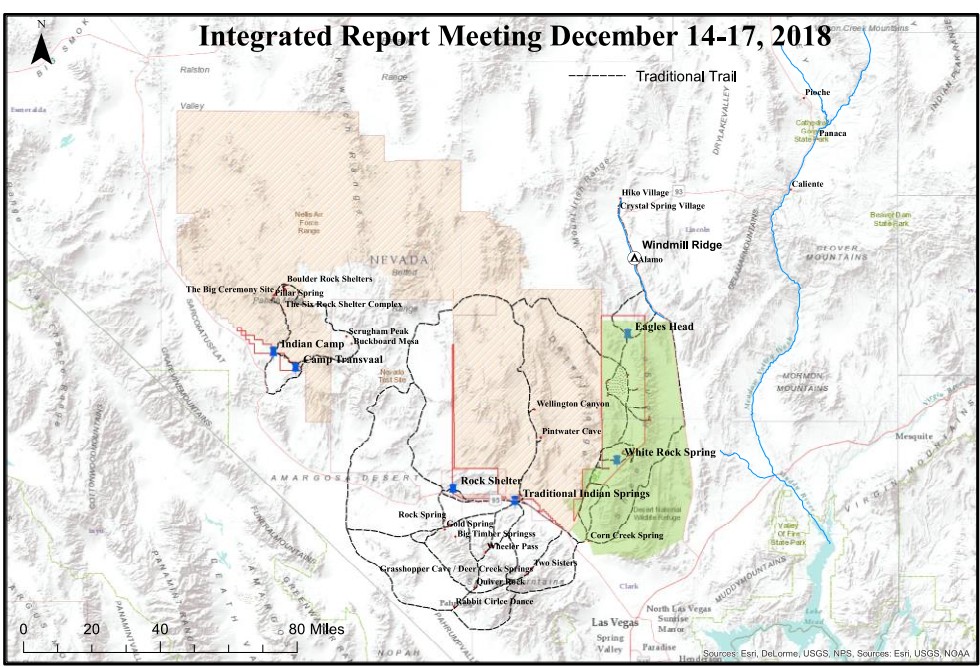

**Figure 3.** Federal agency boundaries and documented traditional trails [40].

## 4. Methods

The LEIS ethnographic study was prepared by a research team from the University of Arizona, School of Anthropology who were funded by the U.S. Air Force to facilitate the cultural understanding and assessments of 18 formally consulting tribes, who referred to themselves as the Consolidated Group of Tribes and Organizations (CGTO) [48]. The UofA research team was specifically asked by the CGTO to conduct the study based on past research relationships. The study findings were reviewed for accuracy by the members of the CGTO and members of the CGTO Writers Committee before being released to the LEIS writing team and published in the Final LEIS [35].

The LEIS Study began with a Scoping Trip involving the CGTO Writers Committee and a DNWR archaeologist. This trip was conducted in late September 2017. After visiting a number of locations, it was decided that the field studies would include (1) Black Butte in Pahranagat Valley, a spiritual place, (2) Eagle Head, a known archaeology site, (3) the Desert Lake Playa, a prominent topographic feature, (4) White Rock Spring, a known archaeology site which combines archaeology, a spring, and an ecological shift to an upland ecology, (5) the Joshua Tree Forest, which dominates the ecology of the lower valley, and (6) Corn Creek, a pilgrimage support community. Sites #1 and #6 were informal interviews because they were important to the overall understanding of the cultural landscape but located just outside of the 3C boundary. All locations were identified as components of a cultural landscape and a pilgrimage trail.

This analysis includes *tiering* information, which situated and elaborated on the cultural meaning of resources and places that had been identified in earlier studies by the CGTO Writers Committee or other Native research teams. All tiering information derived from past cultural studies conducted near the LEIS study area and was, thus, considered to be directly relevant. Many are referenced in this analysis. Findings from previous studies

document the cultural meanings of the Pahranagat Valley, Black Butte, Corn Creek, and the Spring Mountains. The conclusions from these previous studies had all been approved at the time of the study by the participating Native American representatives and their consulting tribal governments.

The field portion of the LEIS study in area 3C began in December of 2018 in a large conference room at the Corn Creek Headquarters for the DNWR (Figure 4) [37]. At this orientation meeting, the CGTO Writers Committee representatives received a welcome by the DNWR Superintendent, a visit to the interpretative museum, a presentation by the DNWR archaeologist, and a hard-back field notebook containing maps and descriptions of each site to be visited. In the notebook were Data Collection forms specially designed to be filled out in writing by each member of the Writers Committee at study sites. A form designed to organize potential observations was provided for each site. It contained resource topics that had been approved by Native representatives in previous studies. In addition, each member of the Writers Committee was provided with a voice recorder that could be used to document their thoughts while walking to and from the study area sites. Individual observations, both written and recorded, were supplemented by an end of day circle discussion by the whole Writers Committee. These circle discussions were open ended but focused on collective observations.

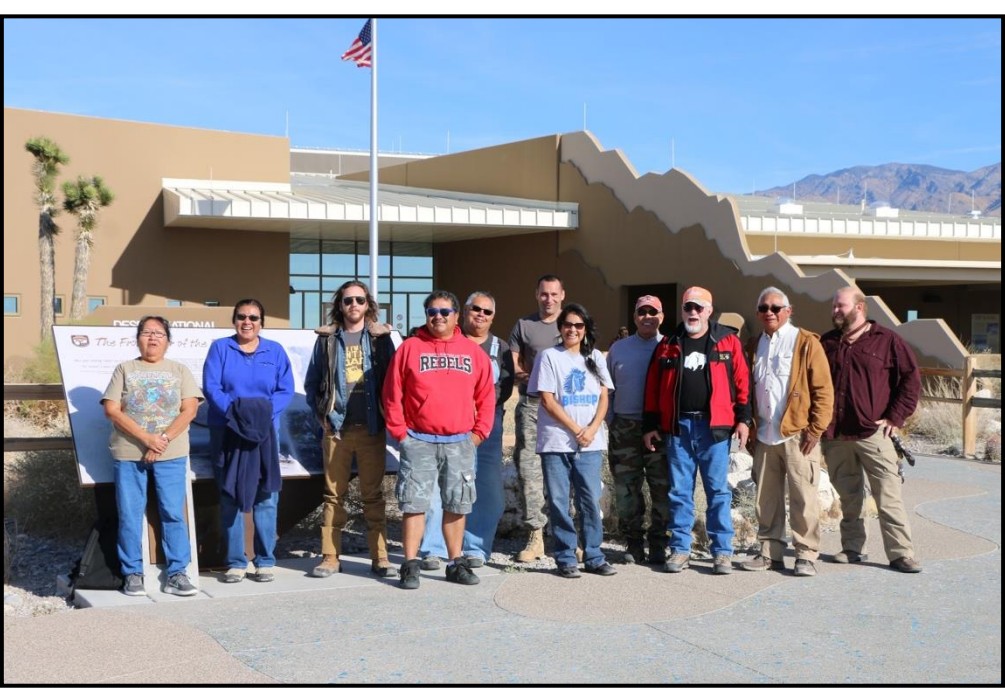

**Figure 4.** The 2017 3C Field Team included three UofA ethnographers and eight members of the CGTO Writers Committee at Corn Creek Headquarters for the DNWR (Source: Stoffle).

The Writers' Committee was appointed by the 18 consulting tribes to review environmental and cultural documents and conduct field research on their behalf. This CGTO LEIS Writers Committees had worked together for almost a decade, so they were both experienced and had extensive cultural background.

## 5. The Cultural Landscape

The notion that the land and its many resources are integrated and have been designed that way by Creation itself is basic to Native American interpretations of all the Earth [49–51]. The following discussion of the integrated cultural landscape was offered by a tribal representative while he looked west from White Rock Spring at the mountains and the playa in the valley below (Figure 5):

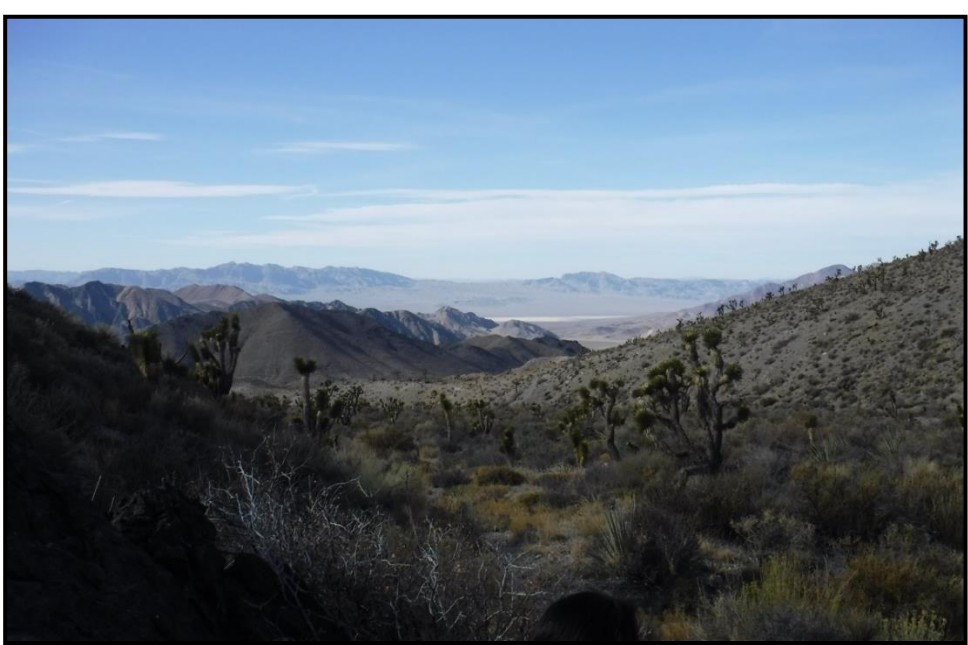

**Figure 5.** A Cultural Landscape View West of White Rock Spring (Source: Stoffle).

*The first thing I saw when we started coming up from the valley to White Rock spring is you have this very vast canyon. And this is exactly where it is bound by mountains on both sides that help resonate songs or voices from Mt. Charleston that are coming up this way. The mountains are like repeaters. It is part of those echoes, part of those voices coming up the valley. The songs are also kept within these mountains. This big valley comes all the way up and heads up north, there are some mountain peaks in there that are talking to some of the old folks. Some of these mountains were central to helping the songs and the beliefs keep this area in balance. And I think that is really important. When you look at this Joshua Tree Forest it is probably a good ten miles long and four miles wide. It is vast. This makes up part of the cultural aesthetic, part of the landscape, that tribal people know. Just as the roasting pits that are around showing that we came to these areas and everything that you see here was put here for a reason. There is not only food, but also medicine too.*

*This landscape is like a basket. So, when you put everything together, and when everything is all woven together it is strong, a strong basket. And it holds that strength. This valley is the central corridor. It is just like that with the songs we have. A lot of the time people rely on the Salt Songs that we have, and those are very, very important, the songs for our journey [to the afterlife], but there are other songs. There are the Fox Songs, Silver Songs, Mountain Sheep Songs, Turtle Songs, and the Badger Songs. These songs go on and on. And there are songs that have to do with the sky and keeping everything in balance. So, all of those songs come up this valley and interact with everything else the mountains, the water, the plants, the animals, and the wind. These songs act almost like a blanket that goes over all of this. That is what holds it and keeps it intact. And then when the land is needed, you call upon it as you're walking, and it will come to you.*

*This cultural landscape is what is vital to us as a people. This is what we have seen. Those mountains are still the same as when the people were here a long time ago, when the world was new. Those mountains heard that, saw that, know what is going on. We are right here seeing those exact same things that the other people that were up here saw. They looked in that direction and saw exactly what we are seeing. Although some of the plants and the landscape has changed: there was more water around, and that is in our stories. The playa lake was full of water. The plant communities were different. That is because the weather changes.*

*The songs are part of the vocal snapshots of the land. In those songs they describe what this land is, what it means, what it is supposed to look like, how it is supposed to be cared for, and how you interact with the land. It is like the rules, the rules of engagement, your words that you are singing to all of these things. So, we are saying how lush this is, we are talking about those forests, these Joshua Tree forests. That is, when it hears us, what it remembers. And then, so that is what it wants. It is trying to help hold onto that stuff, hold onto the way it is supposed to be. In songs you are describing the area. You are showing your appreciation for it.*

*This is probably one of the most unique valleys just because there are no distractions here. This is very, very important. You can see straight up there, there is Mt. Charleston, it is there, that is the place where we were all Created, this is part of the areas that the old people would go. This is how we got into other areas. We came right through here. Those songs too come right through here. The people and songs have been here, and the land has held on to those songs. You cannot find a more pristine valley.*

*5.1. Pahranagat Valley*

There is a dark volcanic butte in the middle of an extensive riverine oasis filled with clear fast flowing artesian springs, creeks, lakes, and marshes (Figure 6). This is Pahranagat Valley, and it was the Time Immemorial traditional home of large irrigated agricultural villages, abundant fishing, and extensive plant-gathering areas. The people of the valley and those along the Muddy River downstream were related as a single district and, together, they had their own origin place called Coyote's Jar, a peak in the mountains surrounding the valley. At the narrow, southern end of the valley is an old Pleistocene Lake bordered by a 15-foot tall, continuous white band at the base of the cliffs caused by the lake filling and partially drying out in the past. This is the Origin place for the redtail hawk who sat on the wall of the cliff, thus letting his tail dip in the lake. He thus acquired a white band across his tail.

The White River flows through the Pahranagat Valley; thus, it is central to an extensive hydrological system that feeds into the Virgin River and subsequently into the Colorado River to the southeast. The area is topographically a part of the Colorado Plateau, but it is adjacent to the Great Basin. The Pahranagat hydrological system has permanent and abundant water because it is fed by snow and rain that fall on much higher surrounding mountains, especially near its headwaters to the north. While the mountains are covered with dense forests, the valley itself is arid. Together, these produce a fertile riverine oasis, which has supported Native American agriculture and life since Time Immemorial.

A similar environmental perspective was recorded in 1864 when William Nye and his mine explorers lived a winter in the valley. In his essay entitled "A Winter Among the Paiutes," Nye [53] noted that:

Pah-ranagat is purely an Indian name, and one which in the Piutes dialect signifies "shinining water"—the Valley of the Shining Water—a name which, at least, reflects no little poetic faculty of the Indian dwellers in this valley of the mountains. After all, it is a pleasant thought, that in the past that little strip of fertility with its grass-bordered streams has been an Indian paradise.

Nye's mining camp was located on a mountainside within view of Lower Pahranagat Lake. Below the camp was an Indian village whose inhabitants grew corn and melons. The Fowler and Sharrock [54] and Madsen studies of this valley, based on field archaeology and documents, indicated that from the protohistoric period until at least 1865, the "Southern Paiutes were farming with the use of irrigation ditches in the valley." It probably was the location of a High Chief [55].

Although Nye [53] noted the common belief of his fellow Americans that Native Indians and the White Man are natural born enemies and that we are here only to fight and kill each other, it was Nye's experience that:

Ours was an honest struggle to live in peace with our Indian neighbors; and we found them, in many respects, not very unlike what any community of two hundred white men would have been under the same circumstances . . . Their chief (Pah-Wichit) saw fit, at the outset, to remind us that the region was his domain. He said: "Me one great captain," and with impressive gesture, he pointed down the valley . . .

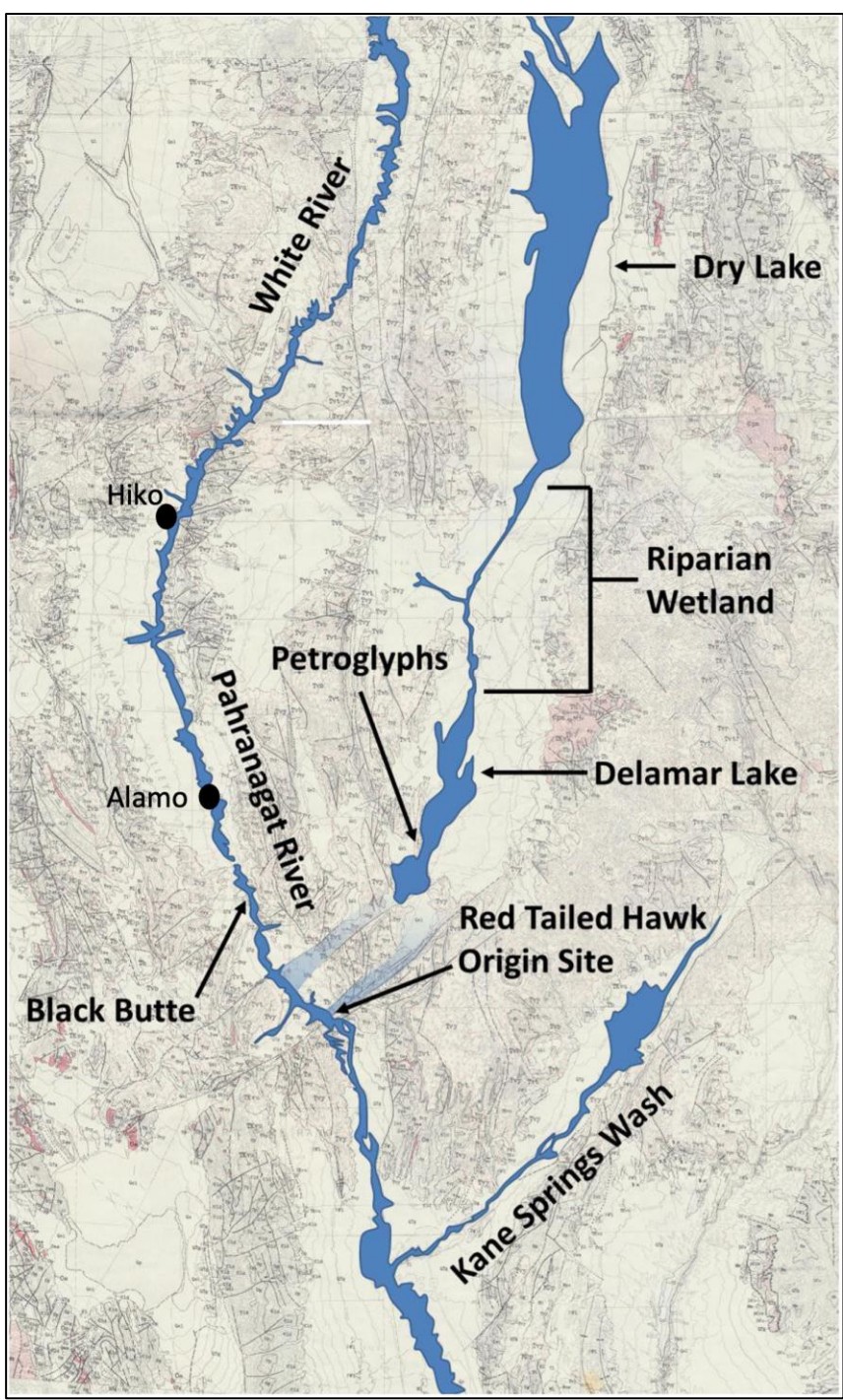

**Figure 6.** Hydrological systems in Pahranagat Valley [52].

Despite the early efforts of Nye to forge a peaceful coexistence with the Indian villages of the valley, the State of Nevada was to establish the first county seat just upstream at a community called Hiko. Before that could happen, however, there had to be a growth

in local population so new settlers were encouraged to mine and farm. As a matter of Nevada Territorial policy and of image, the Indians had to be controlled. In the late 1860s, the members of two of the three Indian villages were surrounded and most of the people were either killed or removed.

In 1873, Powell and Ingalls recorded a Native population in their survey. At that time, there were 171 Pahran-i-gats residents in the valley headed by a chief called An-ti-av [56]. The cultural significance of places such as Black Butte in Pahranagat Valley to Native people is well understood from Powell and Ingalls who interviewed the Native American farmers in the valley. During that visit, they recorded a poem that they had composed regarding Native feelings about the beauty of the area. It was a Paiute song about this valley, which they called The Beautiful Valley [MS 831-c] [56].

| | |
|---|---|
| *Pa-ran'-i-gi yu-av'-i* | *The Paranagat Valley* |
| *Yu-av'-in-in* | *The Valley* |
| *Pa-ran'-i'gi yu-avai-I* | *The Paranagat Valley* |
| *Yu-av'-in-in* | *The Valley* |
| *U-ai'-in-in yu-av'-I* | *Is a Beautiful Valley* |
| *Yu-av'-in-in* | *The Valley* |

While this song and poem do reflect the traditional views of the Indian people about this valley, Powell and Ingalls recorded but a small portion of this tradition song. Many songs lasted for hours. This song was composed for and traditionally sung to the valley. According to the Powell and Ingalls report, the Pahran-i-gats were formerly three separate tribes (probably local irrigated farming communities), but their lands had been taken from them by white men and so they have united in one tribe under An-ti-av [56]. Two years later, in 1875, all local Paiute Indians, including the Pahran-i-gats, were relocated from their valley to the new Moapa Indian Reservation on the Muddy River.

### 5.2. Black Butte

Black Butte is rich in cultural resources. The more obvious feature at this place is a volcanic butte located in the White River in the center of a large riparian wetland (Figure 7). This is a special puha place for rain shamans. The volcanic butte at Black Canyon has the *cultural place logic* of a rain shaman's power place including evidence of volcanic activity, abundant flowing water, a small narrow canyon constricting the water flow, tobacco plants growing out of the cliff faces, and vistas. It is a place of great power in a valley of great power. Along the cliff faces are numerous large and unusual peckings. These seem of great antiquity, but portions have been re-pecked, thus documenting repeated use. The Mountain Sheep images are interpreted by Indian people as symbolic of the spirit helper of the rain shaman. Present also is the symbol of an extremely powerful spiritual being—a water baby. Normally, water babies are not used as spirit helpers because they are so powerful and unpredictable [57]. On this volcanic butte, however, a person who is already a powerful rain shaman can connect with water babies and concentrate great puha for his rainmaking ceremonies. Indian people say that visitors from far away came to this site, because the top of the butte is covered with large stone houses where they stayed. This may be one of the major rain shaman ceremonial places in the region.

The participants in the LEIS fieldwork and two earlier studies [44,58] identified the Black Butte as a *cultural entity*. They also identified plants, animals, water, small flat rocks, and the style of the peckings as having cultural significance.

Isabel Kelly interviewed Paiute people in the early 1930s and documented the importance of water and water trails. The following is a rendering of the Kelly notes by Fowler [59]:

> Water itself is a sacred substance to Southern Paiute people, and it must always be approached as a living thing, which means prayerfully. It has its own spirit, and there may also be other specific spirits that live in springs and other water sources that need to be carefully considered. Some of these can be harmful to humans, and thus they, and their water homes, need to be approached with great caution and respect. Springs are viewed as interconnected, with water in many ways being

like the blood of the earth, flowing in veins under the ground and emerging to the surface only occasionally. The Doctors and other men of power could often travel on these underground trails. Water was their mechanism, and the interconnection of springs their pathways. Water spirits can do the same, although the Old People used to say that they, like people, had preferred home—certain springs that they preferred and where they stayed. People knew where these were and always approached these very cautiously and with the utmost respect.

Petroglyphs completely cover the side walls of Black Butte (Figure 8). Hundreds of petroglyphs ranging from complete panels, covered with inter-related figures and shapes, to peckings on isolated boulders. Critical to the interpretation of the butte as a place that is a component of a cultural landscape are the transformed shaman and water baby peckings (Figure 9). A shaman has spirit helpers and, together, they agree to bring patients, communities, and even the world itself into balance, that is to heal. The shaman during the balancing ceremony opens himself or herself up and a spirit helper that passes from another dimension through the shaman to the patient or community. During the passage, the shaman is described as like an open window, framing the event through song and prayer, but only a partner with the self-activating spirit helper. The healing or balancing cannot be accomplished alone by either. Every shaman has multiple spirit helpers who have come to him over his lifetime and asked if he wanted a co-partner. The shaman can choose the spirit helpers who help with the healing specialties of the shaman. During his or her life, a shaman can seek new songs, prayers, and helpers. This is accomplished at places such as caves, springs, and mountains that have a special concentration of puha and powerful beings which can both educate and help the shaman. Black Butte is such a place.

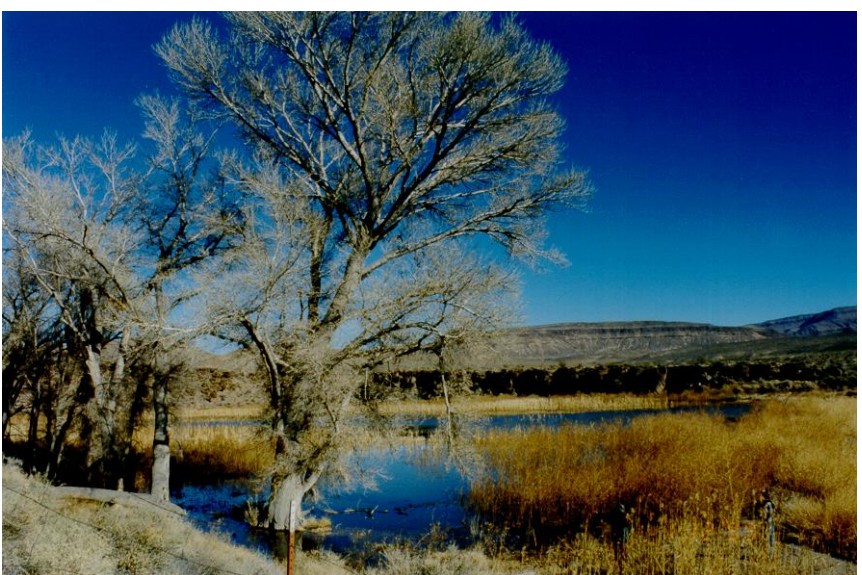

**Figure 7.** Wetland at edge of Black Butte (Source: Stoffle).

Figure 10 presents seven peckings of transformed shaman. Facing outward from one of the sides of the butte, there is an *atlatl* (i.e., a circle with vertical line) or medicine cane in each right hand. The atlatl is the sign of authority being a traditional source of power for hunters. The left hands represent transformed water baby hands. One potentially has medicine cane. Knotted strings, which are what medicine men and community leaders use to convene groups for ceremony, are found both on the surface of the stone face and most commonly occur below the rectangular body. The interior of bodies contains patterns that indicates something about the nature of the transformation. Most representatives interpreted these interior patterns to be culturally meaningful and perhaps related to the kind of shaman being transformed. Note the eyes are natural holes in the cliff face and most likely portals to another dimension. Mountain Sheep peckings are at the lower right.

Mountain Sheep are understood as a spirit helper for rain shaman [60]. This is a pecking pattern that is repeated throughout the Black Butte.

Especially well documented as a sacred trail marker are the knotted string images, called *tapitcapi* (literally "the knotted"), in Paiute. Such strings were sent out by medicine men or community leaders via a runner to people in distant communities to inform them of special events [61]. *Tapitcapi* are similar to *Khipus* (twisted cord) used by the Inca religious elite and leadership in the Andes [62]. Perhaps the best account of these spiritual runner trails is provided by Laird who married one of the last Chemehuevi Paiutes ritual runners [61]. Such runners could move along these trails without time elapsing. The special trails were specifically created. The trails were complex because they passed from water source to water source across the rugged terrain of the Mohave Desert and Colorado Plateau regional landscapes, but they also involved portals so the runner could travel through other dimensions. Often trails were traveled at night. To remember the trail routes, runners would sing a song that told the way. These trail songs described the path to be followed, as well the time it should take, and encouraged the runner by recounting stories of mythic beings who traveled or established the same trail. The trail songs were so critical that ownership was limited to specific individuals and families who maintained the songs and passed them from generation to generation as a heritage [61].

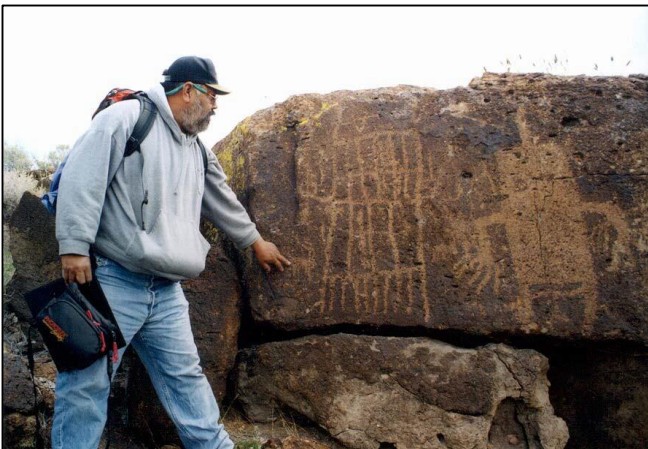

**Figure 8.** Co-researcher and Western Shoshone Elder from CGTO Writers Committee pointing to a mountain sheep pecking while explaining transformed shaman and a water baby to the right (Source: Stoffle).

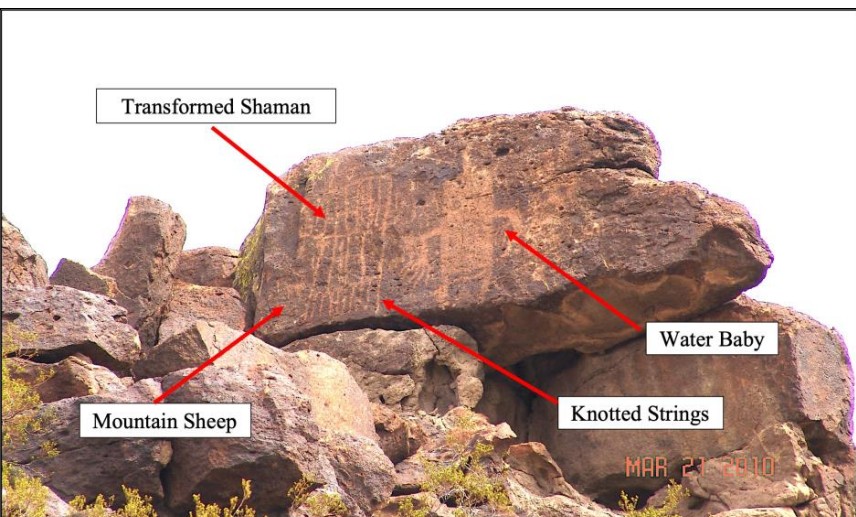

**Figure 9.** Analysis of shaman and water baby patterns from Black Butte (Source: Stoffle).

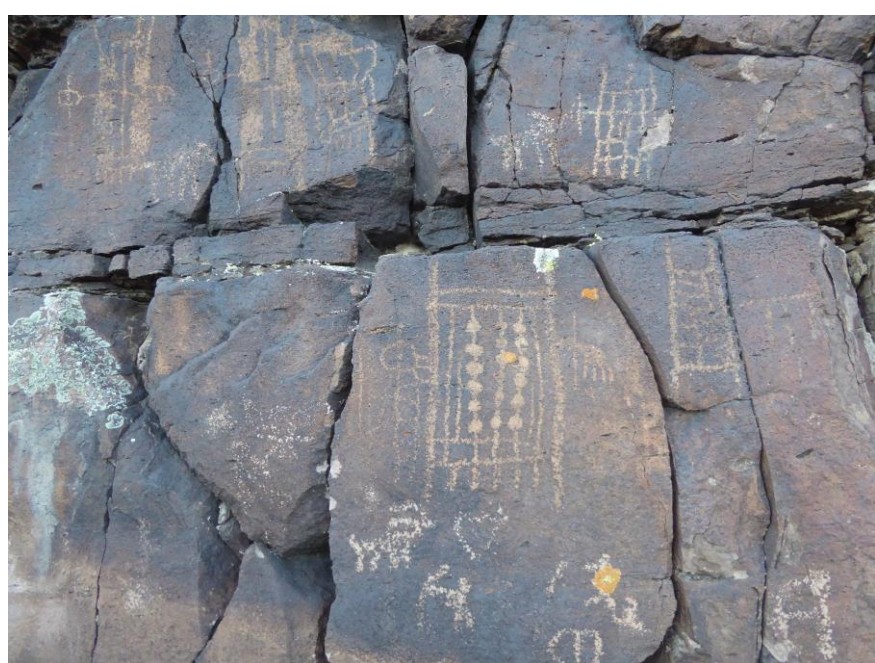

**Figure 10.** Peckings of transformed shaman (Source: Stoffle).

Paiute and Shoshone people in the late 1800s and early 1900s had many kinds of shaman as was recorded by Isabel Kelly between 1932 and 1934 [63]. Laird's husband George [61] explained shamanism:

> In his dreams, the novice acquired a super-natural spirit (*utu'xuxuar; or tut.uyua'*) which means the source of his healing power. This Being took the form of a man, woman, or child; but, more frequently, of an animal such as a bear, coyote, wildcat, lion, eagle, and so forth. According to one informant, the spirit was not visible to the dreamer immediately, but he saw it later, not in a dream, but in the daytime, when alone. Several such spirits might be acquired during a lifetime. They were visible to the shaman only, and although his professional colleagues might know of their identity, the layman ordinarily did not.

Kelly [63] and Laird [61] document the need for a shaman to change and expand through time, thus requiring other power places to go to for more songs, prayers, and helpers. According to Carobeth Laird [61]:

> George invariably translated *puhwaganti* as "doctor," and every genuine doctor was also *tutuguugwanti* as one who has a helper and was presumed to have the ability to heal. *Tutuguuvi* means helper, spirit animal familiar. Those who were shamans in the story-time who became shamans' helpers in this present time. However, not every helper with whom the shaman had contact enabled him to act beneficently at all times. Whether a shaman was consistently helpful or potentially dangerous depends on what sort of familiar he had.

These ethnographic details are essential to this analysis because they document the value of having access to dozens or hundreds of powerful places to interact with other dimensions and acquire new or different spirit helpers.

Water babies continue to be discussed and understood by Native peoples. Several participants gave accounts of their experiences with water babies. They largely agree that Black Butte is a place of water babies, which were discussed in ways that implied a perpetual presence.

> There are water babies there. There are quite a few of them. They can also fly in the air. The people are kind of scared of them; sometimes they get to know them. They get the power of the water babies and when they do this, they get thrilled and just fall down. They

*get married to the water baby—that water baby is going to be mine. 'Tuhumpingang' means to marry the water baby spirit. Water baby is a strange spirit. 'PÔng ipits' is the water baby. It looks like a little child. I have seen the water baby marriage a number of times. It happens to both men and women. It last ten to twenty minutes and after, they get up from their [trance] and are excited and happy, with lots of energy* (Southern Paiute man).

A significant aspect of Black Butte is the combination of water and rock peckings, both of which have power and are connected to the presence of water babies or *PÔng ipits*. These spirits are a recurrent theme among Numic groups from California to Wyoming [64]. Whitley [65] identifies them as shamans' assistants, imparting further emphasis to the shamanistic importance of this place. Pecking figures are beside large cracks or holes in the rock (see image in Figure 8 above), which were interpreted as passages to the supernatural world. These fissures allow the shaman access to this world where (s)he plans to communicate with a water baby. Generally, water babies are considered dangerous, and a shaman must be brave to engage it and return with new knowledge and understanding of the problem for which he is seeking an answer [64].

Figure 11 is a photo of stone walls that were built on the top of Black Butte. The representative's interpretation of these butte walls is especially important for this analysis of landscapes and pilgrimage trails. The dominant interpretation was that these were placed here to provide shelter and a place of prayer for visiting *Puhagants*. Had they just been travelers, they would have stayed with others at one of the three agricultural villages just a few miles upriver. From this location, they would either travel back to their home communities or for additional spiritual learning and acquisition to the Spring Mountains in the south.

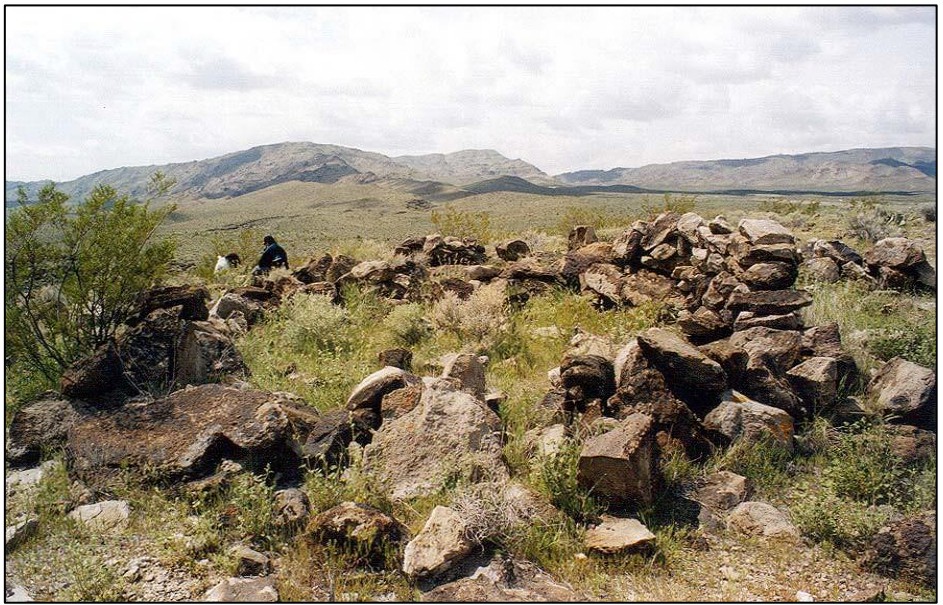

**Figure 11.** Circular stone walls on top of Black Butte with view to Southwest (Source: Stoffle).

Plants are connected to this site through medicine, ceremony, and healing (Western Shoshone man). Medicine plants include Indian tobacco (Figure 12) often grow out of the walls of the Butte. Present also are sacred datura, sage, Indian tea (Southern Paiute woman), and rye grass [58]. Indian tobacco was used also for trade (according to an Owens Valley Paiute woman) and the rye grass was used for food, ceremonies, and making things [58]. Other plants connected with this site include paintbrush, cactus (according to a Western Shoshone man), willow, and sego lily (according to a Southern Paiute woman).

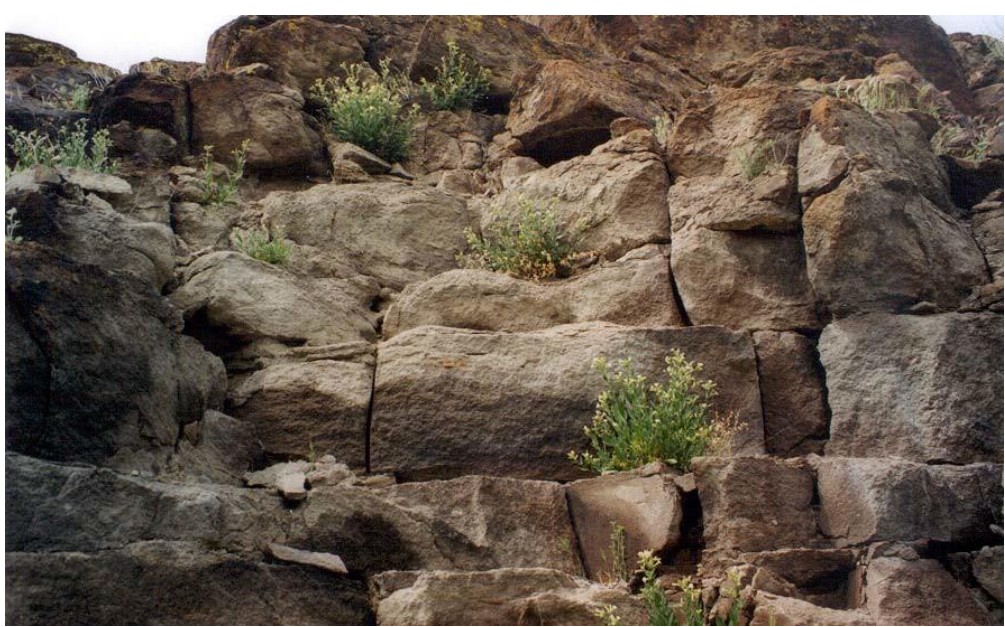

**Figure 12.** Koapi (Indian tobacco, *Nicotiana trigonophylla*) growing out of the Black Butte cliff face (Source: Stoffle).

At any one time, according to a USFWS park ranger, there are 250 species of birds living in the area [44]. The place is on the Pacific Flyway for migratory birds, but it is also the permanent home for many species. A red tail hawk nest was in a large cottonwood (*Populus* spp.) just next to the Black Butte. Mammals in the area include coyote (*Canis latrans*), bobcats (*Felis rufus*), mountain lions (*Felis concolor*), muskrats (*Ondatra zibethicus*), and deer (*Odocoileus hemionus*), some of whom sleep at night on a grassy area just below one of the large cliff face panels high on the butte.

Ceremonial use of plants includes aiding the traditional person's journey to the spirit world and for shamanistic purposes. According to a Southern Paiute woman, a shaman will use a plant that is very powerful and that has led him to it for a specific purpose. Another representative noted that the ceremonial usage of plants continues to be as common today as in the past. Animals have both ceremonial purposes as well as contributing to ceremonial objects such as drums, awls, and prayer feathers, called in the Hopi language *Pahos* [66].

A Southern Paiute elder concluded his interpretation of Black Butte and its relationship to the agricultural villages upriver [58]:

> *My people made these panels. They had a village there and were a very traditional band. They had the opening to the east up there and would keep the young women away from the spiritual men there, the people surrounding the spiritual knoll. They had a farm there and there were more people up the canyon too. The men used this place for ceremonies, to seek knowledge and power, and to communicate with spiritual beings. The men in my family used these panels for these reasons and to communicate with other Indian people* (Southern Paiute man).

A Southern Paiute Woman also noted:

> *As Southern Paiute territory, traditional people from the different bands made these panels and, in this way, it belongs to them. It's understood that these kinds of writings or pecking of panels meant that something sacred occurred and is highly respected.*

A Western Shoshone man stated that "This is a very, very strong, spiritually strong place. It was a meeting place of other spiritual or medicine men".

The tribal study participants who visited Black Butte uniformly expressed a sense of good feeling, comfort, and spiritual strength. In the three studies conducted here, they recognized the importance of the place. It represents Indian control; this is Indian country

at its best. If the land has an ambiance, then true comfort exists here (Southern Paiute Woman). The participants also knew of stories and legends associated with this place. Clear to them it was connected spiritually, physically and through common ceremonial activities to other places throughout the region.

### 5.3. Eagle Head

The pilgrimage trail from Black Butte to Eagle Head is well understood because it was interpreted by the Writers Committee and had previously been identified. The trail exists along where water has flowed and thus mimics where puha moves. It has a clear topographic constriction filled with peckings to mark its presence at Eagle Head and the power of the place (Figure 13).

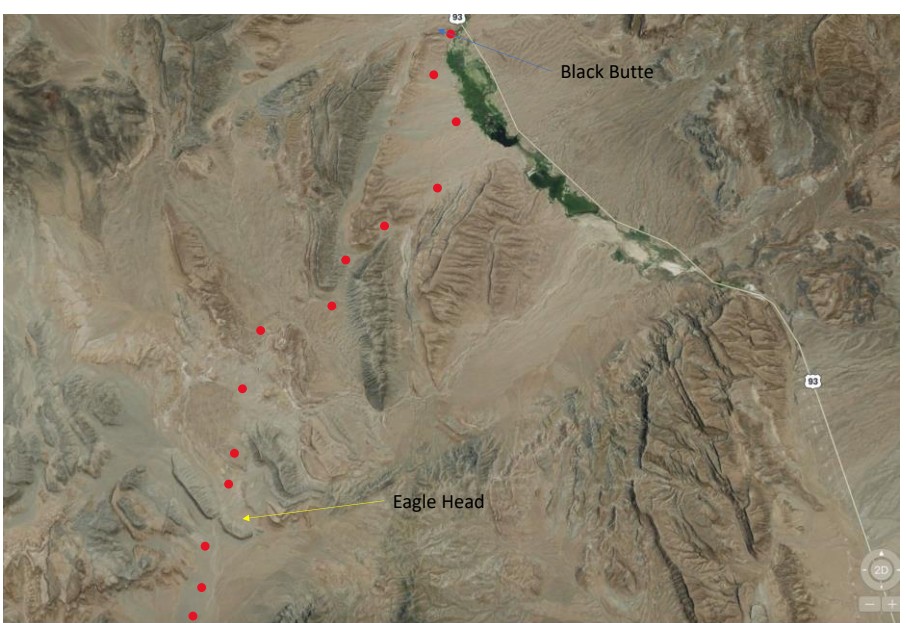

**Figure 13.** Trail from Black Butte to Eagle Head (Source: Google Earth).

Although the Eagle Head constriction marks a trail used by the people who are coming and going, it also represents a place to stop and make offerings . . . a sacred place according to representatives. Offerings, prayer, and ceremony are central to the Eagle Head area. One representative noted that this narrow canyon is a very powerful spot because it is a constriction though which the trail passes (Figure 14).

People gathered Puha here and prayed. The presence of power in this location is tied to the use of Eagle Head during pilgrimages according to representatives. Power can be drawn from a location to perform activities, such as doctoring, which in turn requires prayers and offerings in exchange. The following comments demonstrate tribal representatives' perceptions of Eagle Head as a place for various activities involving power:

- *[This is a] trail with power to keep balance in the world. The rock openings are the voice of the rocks that come back to life when speaking Indian. The rocks hold songs, stories and medicine to protect the land. The location is described in badger and mountain sheep songs used for protection and weather. The badger takes the songs and messages from below to the surface to help Indian doctors.*
- *This location is a powerful place on the journey for "Pahagants" to be doctored and to keep the world in balance.*
- *When people travel, they would give offering to the land for a safe trip.*

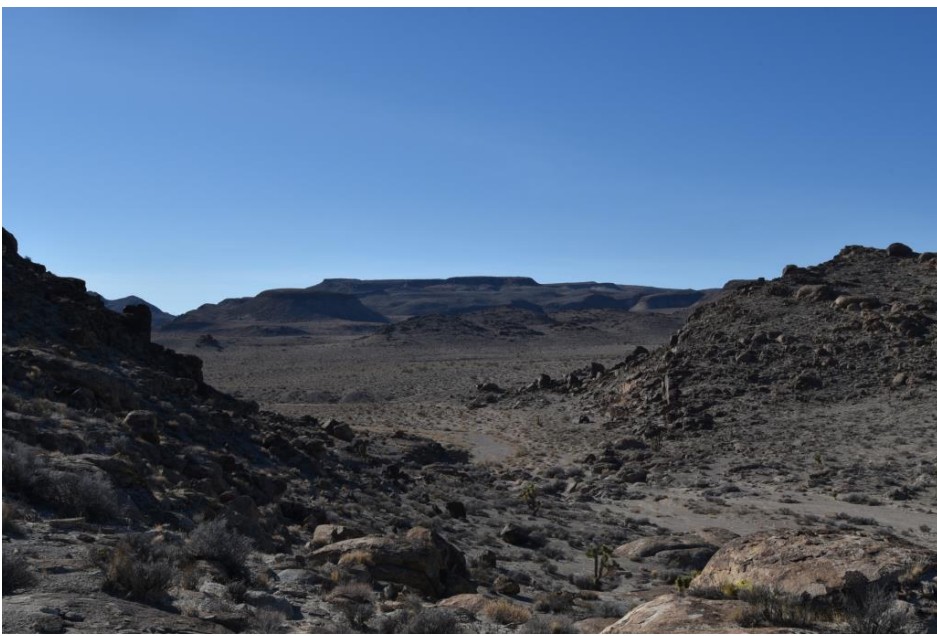

**Figure 14.** Gap at Eagle Head with trail in center (Source: Stoffle).

Nearby is a large boulder at the foot of a cliff, high above the canyon (Figure 15). Here, doctoring was conducted using the power and isolation of the place according to representatives. The rear of the space below the boulder is walled up with large stones and next to it is a flat area covered with grinding slicks where medicine was made.

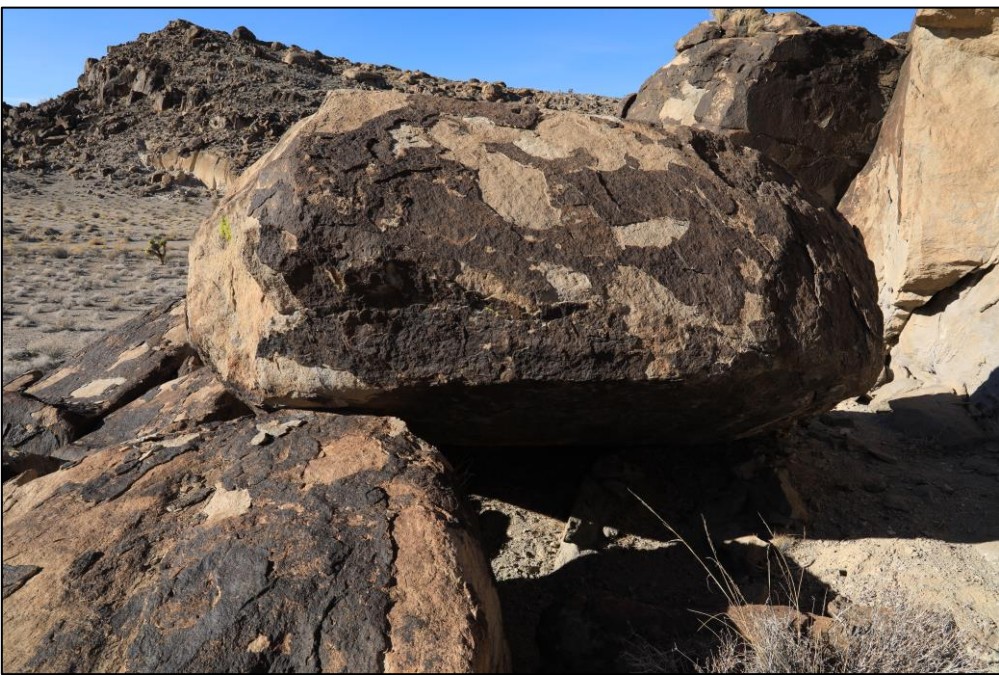

**Figure 15.** A large boulder against a cliff above the canyon (Source: Stoffle).

In addition to describing the area and its use, tribal representatives described how Eagle Head is connected to other places. In accordance with their beliefs, all things are connected which includes people (both humans and animals), places, and things. One representative noted the connection as a *Stream of Puha* that travels to other significant locations. Rain shaman would have come here as indicated by the many peckings of Mountain Sheep, circular origin symbols, and knotted strings (Figure 16).

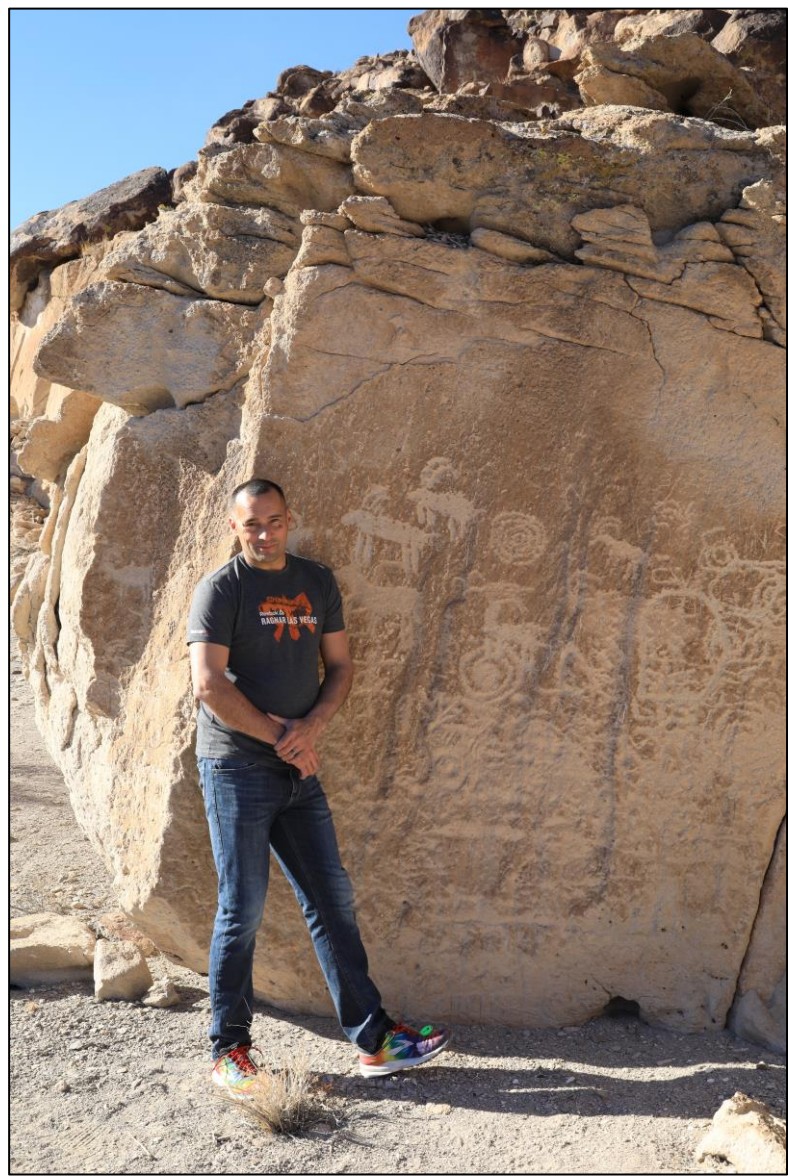

**Figure 16.** Representative interprets pecking panel in canyon (Source: Stoffle).

Tribal representatives noted that Eagle Head:

is connected from the Puha that travels from Black Butte in Pahranagat to Moapa Arrow Canyon, Potato Woman, Sheep Mountains and Spring Mountains. The water carries the message through water babies that come to life at the playas and disappear to go underground.

Water Babies are powerful spiritual beings that live within these areas and are typically associated with peckings found at Eagle Head. Therefore, natural elements (water) and spiritual beings (Water Babies) are both important components for carrying Puha back and forth between places. Another representative noted that this area is connected to the living ancestors of this sacred land: "Southern Paiutes to south and east. Shoshones to west and north." The following quotes demonstrate how this place is connected to other spiritually significant places and the ways through which this connection manifests:

- *This is connected to places such as the Black Butte. It is connected to that place with many water babies and other petroglyph places in Pahranagat Valley.*
- *The rock peckings are located in a short narrow canyon along with hundreds more. Taken together they represent a significant portion of the archaeology found at Eagle Head. All of the*

> *Native American representatives talked about the cultural significance of the peckings and this special place where they were placed. They also talked about who made the peckings.*
> - *Two of the things we need to consider are the acoustics that you hear and the vistas or the views that you see. I think are very, very important. The viewscapes and the songscapes. We talk about spirituality, and we talk about ceremonial activities. But it is almost like we don't talk enough about what we hear and see at a location and how these influenced those old people when they picked places for special activities and events. The acoustics are very apparent here.*

### 5.4. Desert Lake

Desert Lake is the next place studied along the Pilgrimage Trail. It is a playa that, from time to time, partially fills with water but, during the last glacial period, it was a deep lake. This area was observed and discussed from the various places at Eagle Head (Figure 17). Playas, especially those that periodically become dry, are poorly understood by federal and state agencies because there has been a persistent federal agency theory that these topographic features are marginal to Native American life [67]. In general, Indian people recognize that these are old traditional living areas going back to the end of the Pleistocene when the lake basins were full and surrounded by lush vegetation. Slowly, the ancient lakes receded but Indian people continued to live along their shores. For more than 23,000 years [68] and as much as 37,000 years [69] Indian people have lived along the edges of these lakes. They remember those past times and return to the playas today to reconnect with their ancestors and the places that made them strong.

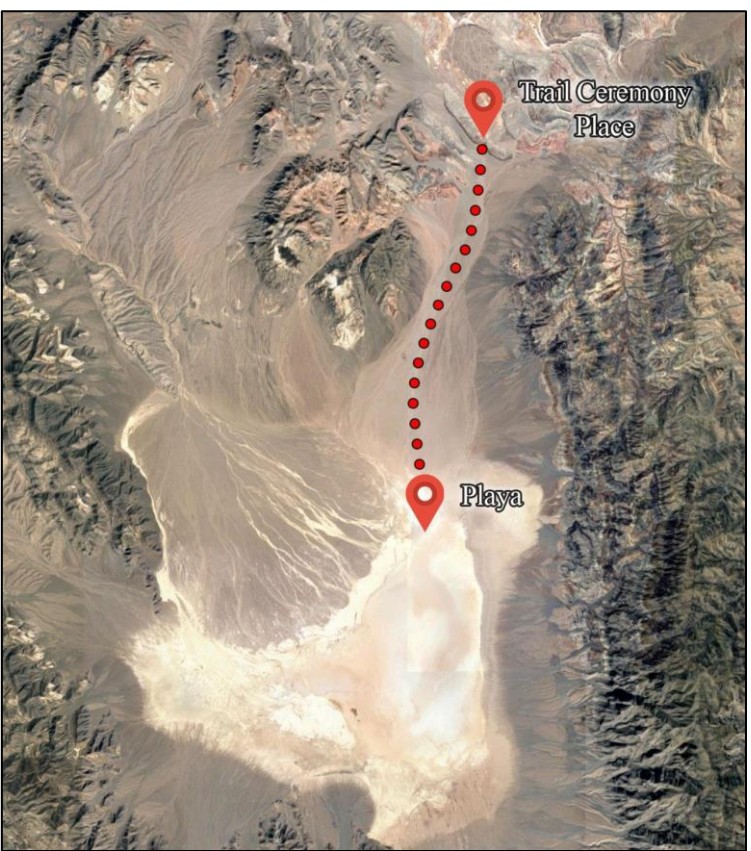

**Figure 17.** Desert Lake Playa (Source: Google Earth).

Desert Lake Playa has been studied by the Far Western archaeology group [70]. A proposal to improve the refuge road that crosses a portion of the Desert Dry Lake playa created a project area involving 322 acres of archaeology inventory and three backhoe trench excavations. In just this small portion of the total playa area, seven archaeological sites were recorded, two of which were recommended as eligible for the National Register

of Historic Places. The study documents that the area was occupied during the Middle Archaic, Late Archaic, and Late Ceramic periods. Carbon dating suggests occupation dates of from 6970 Before Present (BP) to 2000 BP.

*5.5. Joshua Tree Forest and Roasting Pits*

Joshua trees (*Yucca brevifolia*) are one of the major cultural resources that grow in the valley. They are concentrated along the pilgrimage trail in what is called the Joshua Tree Forest where it grows north of the Hidden Forest Road and extends for miles in all directions (Figure 18).

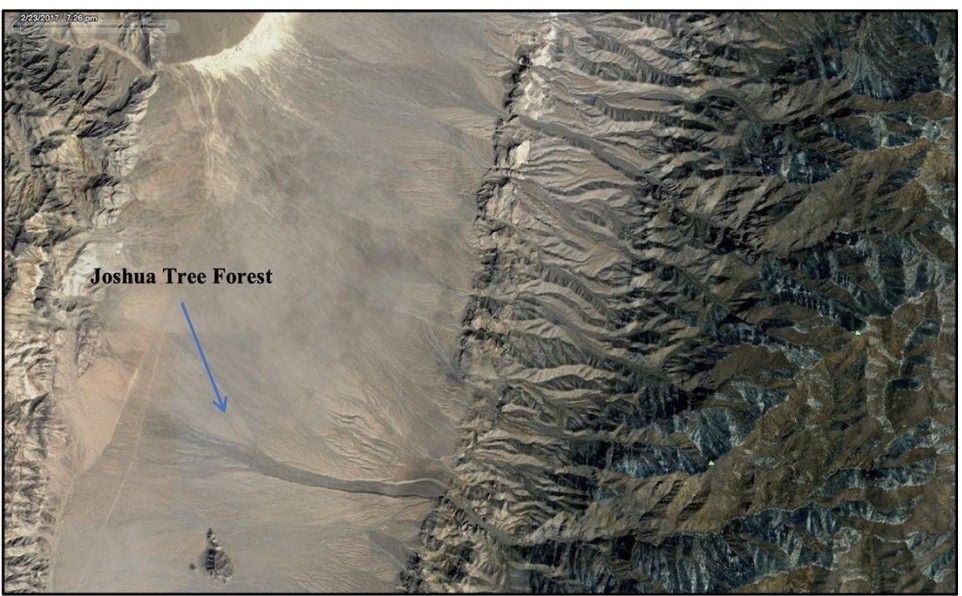

**Figure 18.** Location of Joshua Tree Forest (Source: Google Earth).

Joshua trees (*Yucca brevifolia*), or *umpu* in Western Shoshone, are an important ethnobotanical resource that grow in low to mid-elevation ecological zones (Figure 19). The plant serves multiple traditional functions for Native American peoples of the southwest. Fruits are harvested and broiled or roasted before consumption [71,72].

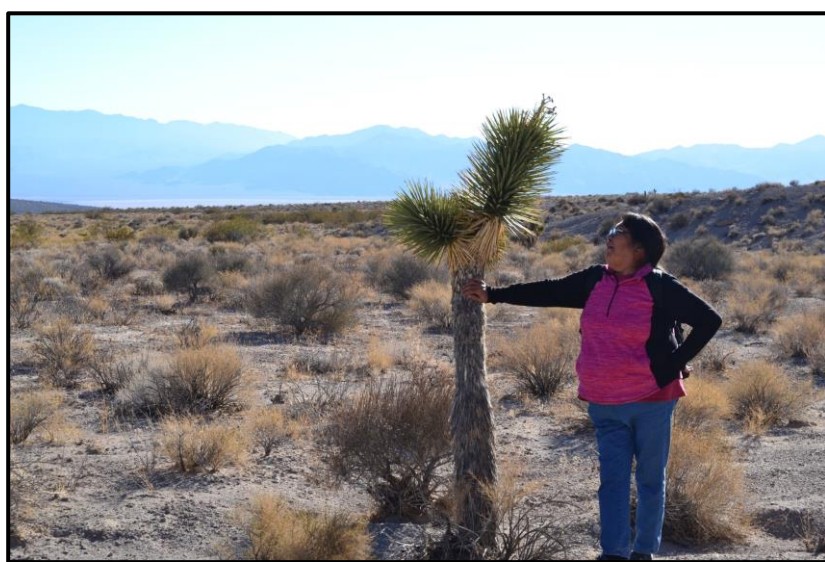

**Figure 19.** Joshua Tree and Writers Committee Member (Source Stoffle).

Yuccas (*brevifolia* and *schindigera*) and agaves (*deseriti*) were a year-round staple producing several types of traditional foods for all Native peoples of the Mojave Desert and Colorado Plateau [73,74]. The flower buds, blossoms, seeds and basal rosette were eaten. Hummingbirds, butterflies, moths lived on this plant, larval of the latter was considered a special treat. Native peoples of the Mojave Desert pounded the leaves to expose fibers which after drying were made into cordage, used in bowstrings, bags, snares, slings, nets, cradles mats, and clothing. The spines were used as needles for both sewing, tattooing, for separating fibers when making baskets. With a handle the needles were used as awls. The roots were also used in basketry and fibers were used in making sandals.

Like many southwest plants, there is some indication that the Joshua Tree was transplanted to other areas to increase availability for traditional purposes. The representatives were especially concerned for the forest given its unique density. They suggested that the Joshua trees were stimulated to grow here. Shipek [75] documents for the Kumeyaay of southern California, for example, that the seeds of agave and yucca were saved and planted in many locations. The seeds were planted immediately before burning a slope, and germination was induced by the heat of fire. These stimulated plants did not provide immediate food but needed several years to mature to a usable size, providing evidence of long-range plant husbandry planning [75].

Agave, yuccas, and other plants with massive fruits are prepared in roasting pits (Figure 20). The roasting pits found within the NDWR remain important to Native Americans in their songs and stories, thus demonstrating cultural continuity. Tribal representatives discussed the construction, use, and cultural significance of roasting pits:

- *The roast was not just limited to agave. That may be one of the things in case they start finding other things around there. We use those kinds of elements that were important. Moreover, they are embedded in songs and stories that we have about the pits, the significance, the rocks that have been brought in for that. That is why songs are not taken away, they have to stay there.*
- *Roasting was communal, and people came from many different areas. And the pit was not just used one time, it was used over and over. So there is continuity in use.*

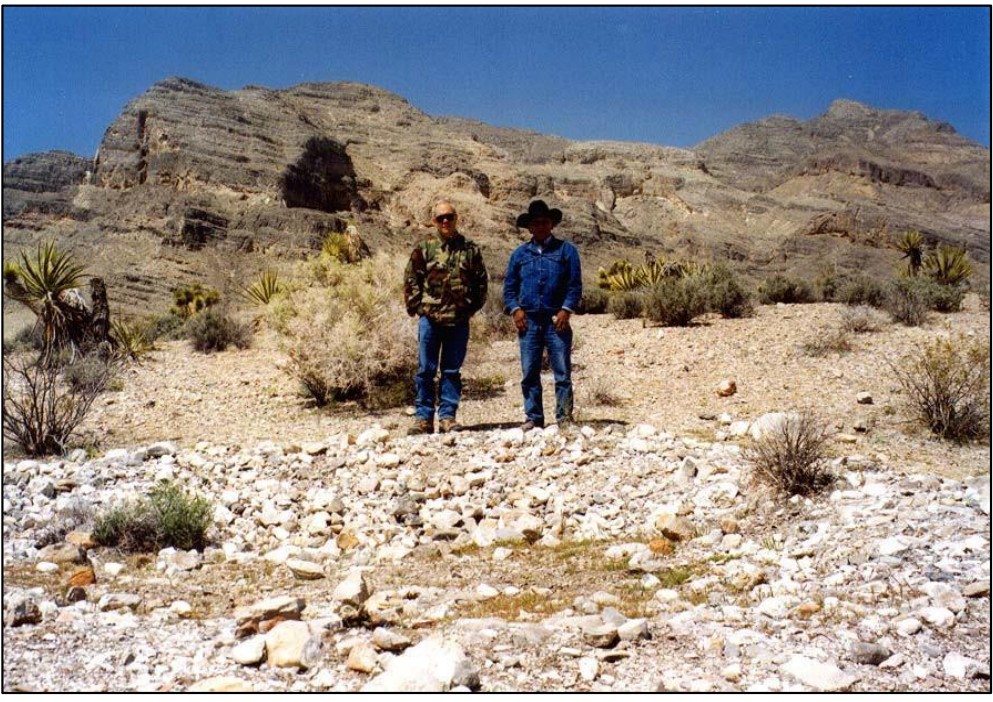

**Figure 20.** Representatives and a large roasting pit (Source: Stoffle).

The representatives were astonished by the number of roasting pits found within the NDWR. In this valley, roasting was done on a large scale with multiple communities

involved; therefore, the high concentration of roasting pits at this location suggests that it was a hub for multiple communities to come together for a special occasion. The following tribal representatives' quotes were regarding the high concentration of roasting pits in this area:

- *It is like a communal event, for one. Roasting activities were not just one person, come in there to cook a hot dog kind of thing. This was . . . everybody came together. The second thing is that we know that it was a communal event, it was conducted with special ceremonies related to the roasting activities. They were conducted by, facilitated by people that were born in summer months, the ones that could do it. They would have their helpers or their assistants to do it. They would enter into the area from the eastern side of the pit to get in there and conduct the roasting activities . . . I think we need to emphasize the interconnectedness between the roasting pits.*

- *It would sort of be like when pine nuts are harvested, and there was planning here which the people came together. People would all come together and roast. It would be like that; people coming together.*

- *This place is part of a balancing ceremony; place where people could come to help restore and rebalance. So maybe this was a hub for those kinds of activities. There are other places around where there are hubs doing other things.*

- *This is where everybody comes together but it is not only the physical part of eating, there is also the spiritual side; connecting us with the land. We are taking something from the land, putting it into us, we are also giving back to the land through the ceremonies, through the songs, through the stories. The third part is, with anything that we do, is also to help sustain that balance and make sure it reoccurs.*

- *You know, it seems like all of those, what you call the Indian Chiefs, they were down there. And they would talk about political things that were going on in the country. They would talk about what is disrupting their lives and how they are going to handle that. That is why these gatherings lasted more than a week. They were not just a couple of days, they would stretch it on for a long time, one or two weeks, because there were a lot of things they had to discuss and talk about and decide on.*

The Joshua Tree Forest is one of few areas in the world where these trees proliferate. Representatives remarked on the importance of this area for this ethnobotanical resource in this special habitat. The Joshua tree is important as a food source and preparing them is a major social event, but the trees provide a key habitat for other important animals and plants. One person noted that animals live in the Joshua Tree Forest, and any disturbance could lead to the destruction of the forest and, thus, many animals would die or leave the valley.

The western flank of the Sheep Mountains has dozens of springs, which emerge due to water from snow and rain. These springs have been used since Time Immemorial by Native Americans pilgrims, but at various times, the water they provided caused them to be the location of massive roasting events.

It remains a debate by archaeologists as to what was roasted, such as the fruit from agave, Joshua Trees (Figure 21), or other plants, but clearly from the deep large rock pits, there was considerable roasting over long periods. Other plants also had massive fruits such as the Yucca Tree (Yucca *schidigera*) (Figure 22) and bloomed at different times.

Because agave does not grow in the valley today, many tribal representatives felt that the abundance of the Joshua trees (Yucca *brevifolia*) and other kinds of yuccas (Yucca *schidigera*) suggested that early stem buds and fruits of these plants were roasted here. It is important to note that the valley was utilized for roasting for thousands of years, during which time the distribution and variety of plants growing there would have changed.

*5.6. Corn Creek*

Corn Creek is an area of naturally occurring artisan springs that derive from rain and snow in the surrounding mountains, including the Sheep Mountains to the north and the Spring Mountains to the south (Figure 23). It is the area of Creation for Southern Paiutes where the Origin Burden Basket came open and all humanity escaped. The Southern Paiutes were last to emerge from the origin basket and so they inherited and remained in these Creation Lands.

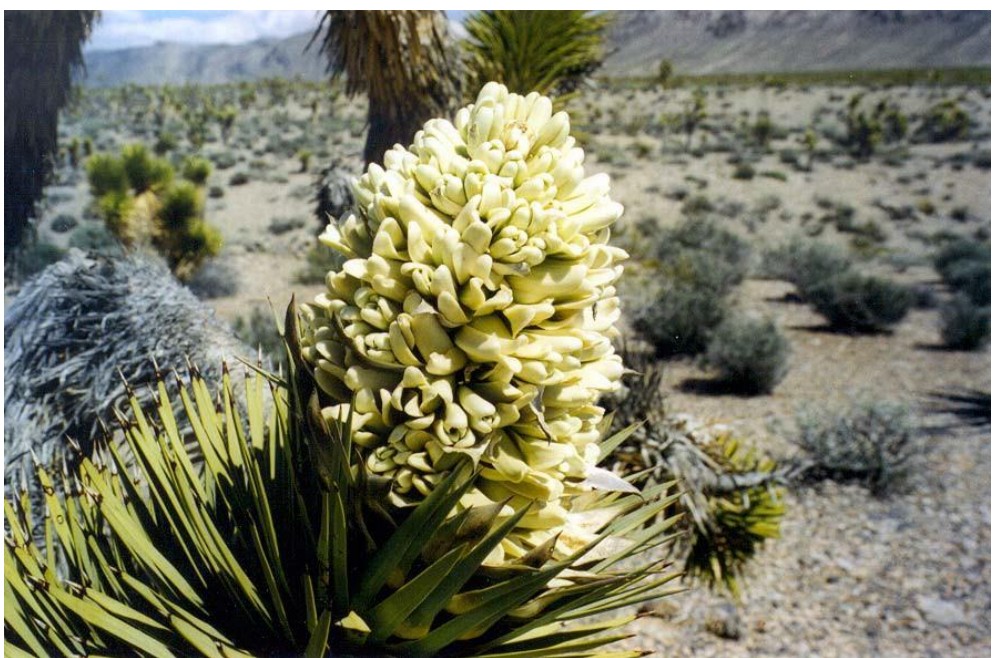

**Figure 21.** Joshua Tree (Yucca *brevifolia*) flower and fruit (Source: Stoffle).

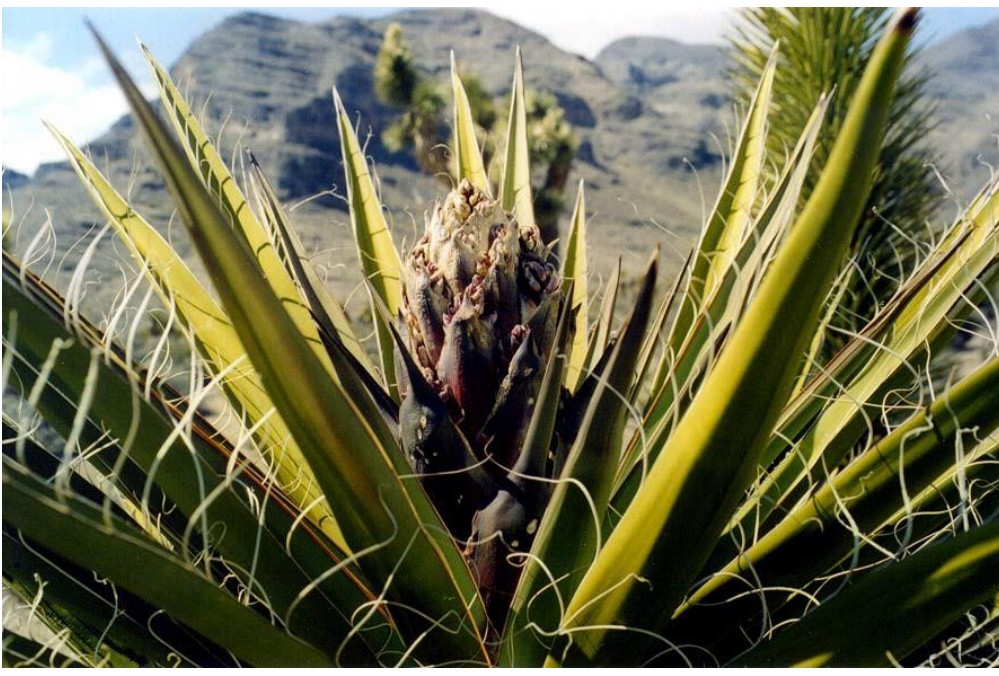

**Figure 22.** Yucca Tree (Yucca *schidigera*) flower and fruit (Source: Stoffle).

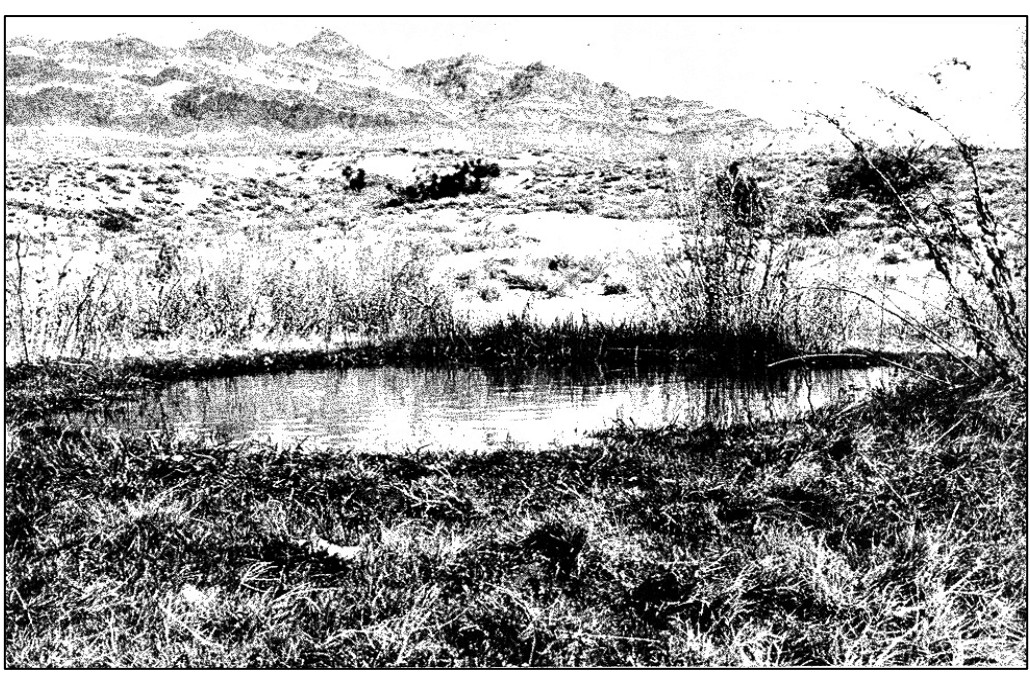

**Figure 23.** Corn Creek Spring with active flow in 1967 [76].

Corn Creek is also an area where Numic speaking peoples farmed for thousands of years [77] at this set of springs and the massive artesian springs in the Las Vegas Valley. They farmed along the Las Vegas River, which still flows to the Colorado River. The river had at least four large, irrigated farming villages located downstream from Las Vegas.

Extensive ethnographic interviews were conducted at Corn Creek during the *East of Nellis* study, and this is a summary of those findings [44]:

> Corn Creek is an oasis spring like those in Ash Meadows. The archaeology of the place documents thousands of years of occupation, with agriculture being here in the proto-historic period. It probably never was the residence of a high chief because it is proximal to such residencies in the Las Vegas and Pahrump areas.

Today, Corn Creek has many trees and other plants including yerba mansa, Indian tea, four- wing saltbush, creosote bush (*Larrea tridentata*), a small pepper plant, mesquite, cattails, cottonwoods, a sweet cane reed, Indian grapes, and sacred datura or *mo'momp*. The cattails, grapes, and cane provided food, while the other species served a variety of uses including for medicinal needs. The *mo'momp*, for example, is a medicine for spiritual beings, "for the guys who traveled through space" (Southern Paiute woman). The large mesquite trees, with their wide-spread limbs, created temporary windbreaks that Indian people use. Given the Creation centrality of Corn Creek, the animals and plants in the area are especially important to the Native Americans whose ancestors have lived in this area since Time Immemorial.

Corn Creek was a place that primarily served travelers. Being near the place where all Southern Paiutes and all other humans were created, it was a support place for ceremonial pilgrims. It may be considered as a special type of ceremonial support community.

*5.7. Spring Mountains: Nuvagantu (Where Snow Sits)*

Corn Creek was a ceremonial support area for people and groups going into the Spring Mountains, which are covered with portals to other dimensions and viewscapes over the region. As such, Corn Creek hosted many kinds of ceremonies, as is culturally appropriate for the nearest natural water source to the origin area for Southern Paiutes.

Ceremonies related to the Spring Mountains are documented by Incised Stones, which have been found on site by DNWR archaeologists. Such stones were brought from afar by pilgrims to be left where prayers were made and to memorialize the place [66].

There is extensive ethnographic evidence of the number and distribution of pilgrimage trails emanating from artesian spring-fed communities surrounding the Spring Mountains. These trails, however, are not marked on Figure 24 as requested by participating tribes. The trails pass into all the high areas of the Spring Mountains. Some of the destinations are marked on Figure 25 to document the general pattern. A selection of these trails is discussed in the Incised Stones essay [66], which documents the use of prayer stones to memorialize ceremonies that occurred along the pilgrimage trails.

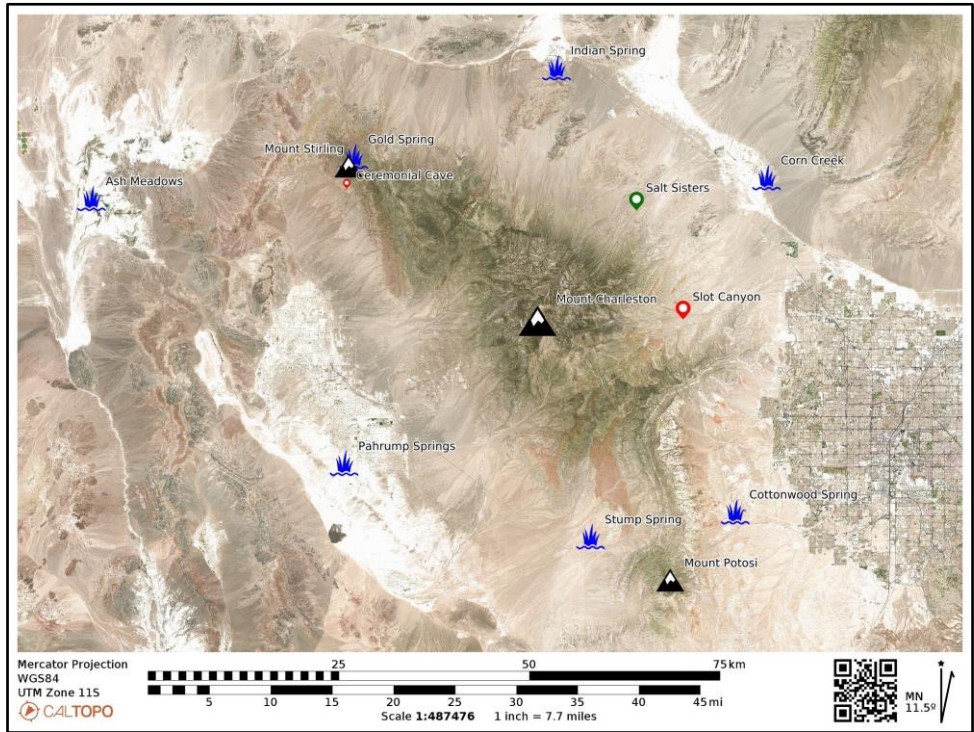

**Figure 24.** Map of Spring Mountains with Pilgrimage Support Communities and Trail Destinations Marked [66].

During the Spring Mountains Cultural Landscape Study [45], Larry Eddy, a religious leader for the Chemehuevi Southern Paiute people (Figure 25), stated:

> *The Spring Mountain range is a powerful area that is centrally located in the lives, history, and minds of Nuwuvi people. The range is a storied land which exists as both physical and mythic reality both simultaneously connected by portals through which humans and other life forms can and do pass back and forth. This is as it was at Creation.*

For this analysis, it is sufficient to note that the Spring Mountains were and are the southern destination of pilgrims who move along the trail from Black Butte and the Pahranagat Valley. Regardless of where their home communities were located, people who move(d) (still occurring) along this trail would pass both ways. Coming to a destination involved the process of mental and physical purification to preparation for a destination and then needing to *clean* themselves of dangerous or just excessive puha on the way back to their homes. Initial cleaning processes involve isolation and hot springs, and the return home cleaning is well documented at natural constrictions along the trail such as with the mazes along the lower Colorado River, the trails to Sugarloaf Mountain at Hoover Dam, the intaglios near Buckboard Mesa, the basalt pecking canyon and caldera exit at Black Mountain, and at the Eagle Head Gap along the Black Butte pilgrimage trail.

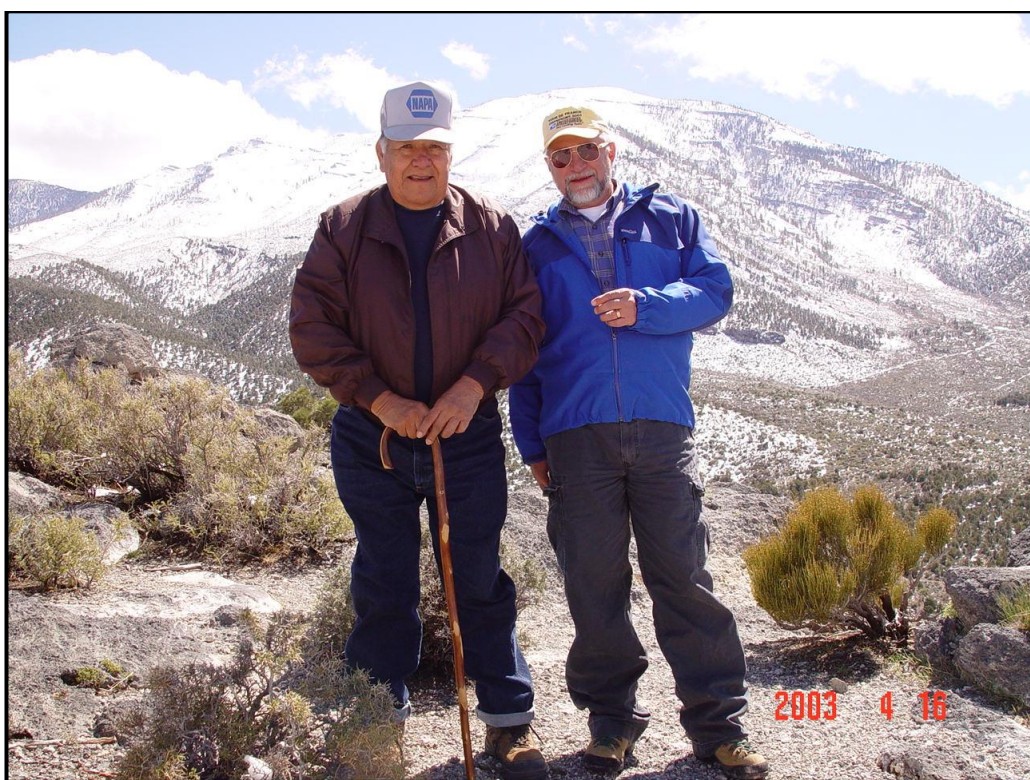

**Figure 25.** Religious Leader Larry Eddy and Rich Stoffle at Spring Mountains [45].

## 6. Discussion

This analysis derives from ethnographic field work and Native American self-reporting in a large valley that is a functionally integrated living cultural landscape. The notion that a landscape and the places within it are alive requires special considerations when human actions potentially impact any of these heritage components. Although beyond the scope of this analysis, living landscapes and component places are expected to occur in the aboriginal lands of traditional people [78]. It is essential to note the concept of *living* derives from an epistemological premise that the Earth is alive and not from the interpretation that it can be considered as alive simply because traditional people still use and care for it [79]. UNESCO maintains that Bali contains one of two living cultural landscapes recognized as World Heritage Sites [80]. While the people of Bali live in and use this landscape, they maintain it is alive [81,82].

The analysis argues that the living heritage cultural landscape presented in this case study broadens the Western academic definition of cultural landscapes. As such, it contributes to the original typology of the ICOMOS cultural landscape working group—designed, evolved, associative—where associative cultural landscapes were originally assumed to apply primarily to aboriginal landscapes without significant physical remains. According to Smith [6], this interpretation is appropriate, but it is being recognized that the category applies equally well to non-aboriginal sites, including ones with a significant built form, when an ecological bias is adopted and the tangible and intangible overlap.

Western research and debate about cultural landscapes should be informed by research from Asian societies, where it is understood that the land and its special places are alive. In Asia this notion has been commonly used in land use, design, and management for thousands of years [2].

Within this cultural landscape valley, and thus contributing to its cultural importance, is an ancient pilgrimage trail composed of living places and spaces. This landscape and heritage trail are the larger cultural frames considered in the eastern portion of Land Withdrawal proposed by the U.S. Nellis Air Force Base and the UTTR. The formal environmental

impact assessment of this proposal on Native American cultural resources is the subject of this field-based analysis.

An environmental impact assessment (EIA) specifies what is impacted in a study area and what those impacts mean to the condition of the resource being considered. Fundamental to this EIA process is agreement that a specific kind of natural or cultural resource does exist and it is present in the Area of Potential Effect (APE) of the proposed action. Since Native American resources have been considered in impact analysis, there has been a challenge regarding the validity of certain kinds of resource identifications because of a fundament difference in beliefs between Native American and most others regarding what is alive and sentient. For Native people, the whole Earth and all its elements are alive; to most others, much of the Earth is inert and not sentient. This has become kind of a problem in environmental communication that is termed an *epistemological divide*. The living cultural landscape that has an interactive pilgrimage trail is such a divide in environmental communication.

Resolution of epistemological divides is essential when the outcomes of environmental communication influences public actions. In a recent book, *Anthropological Perspectives on Environmental Communication* [83], two authors of this analysis wrote about basically impossible cross-cultural heritage discussions: one is about living stone bridges [84] and the other focuses on talking with an active volcanic eruption [28]. Although resolution of epistemological differences is difficult, moving forward in terms of environmental policy and action does occur. The Native American Interaction Program [85,86] at Nellis Air Force Base and the NTTR has had twenty-six years of moving forward with policy and action when confronted with epistemological differences. It is a model of federal recognition of Native American cultural differences and the agency's need to sustain environmental partnerships while maintaining U.S. Congressionally mandated missions.

**Author Contributions:** The authors of this paper each made significant contributions to the data on which it is based and its preparation. The background to the paper involves three decades of research involving university research teams head by R.S. and the 18 tribes who are culturally affiliated with the southern Nevada region. That initial cultural affiliation analysis was prepared by R.S. for the Department of Energy as part of the Yucca Mountain Nuclear Waste Repository proposal [87]. After that identification of cultural affiliation, various federal agencies began government-to-government consultation with the 18 tribes. The university-based research team was funded to facilitate 22 of these ethnographic studies. Data from those tribal and government approved studies have been used for *tiering* the present analysis. R.A. was spokesperson for the Consulting Groups of Tribes and Organizations (CGTO) throughout this period and has been instrumental in designing the research, reviewing the findings, and preparing the reports. Both R.A. and a committee of the CGTO are co-authors of this paper. K.V.V. worked during this time on many of these research projects, was an author of many technical reports and articles, and has been instrumental in the preparation of this manuscript. All authors have read and agreed to the published version of the manuscript.

**Funding:** Faculty salaries were used to support the writing of this paper. No outside funds were provided for the writing of this paper. Funding for the Native American environmental impact assessment was provided by the Nevada Test and Training Range, Nellis Air Force Base for the purpose of conducting a Native American Ethnographic Study for the Legislative Environmental Impact Statement related to proposed land expansions for the NTTR. Those funds were received by the University of Arizona: Contract Number: W912G-14-D-0014; Task Order/Deliverable: DS01 (P010176981); Far Western Job Number: 2007; FRS Account Number: 4021230; The research account was managed by the School of Anthropology at the University of Arizona, Tucson, Arizona after being received by the Arizona Board of Regents.

**Data Availability Statement:** Not applicable.

**Acknowledgments:** The study was co-produced by a UofA School of Anthropology research team headed by Richard Stoffle and the members of the Writers Committee of the Consolidated Groups of Tribes and Organizations including Richard Arnold, Kenny Anderson, Charlie Bulletts, Maurice Churchill, Barbara Durham, Ronald Escobar, Danelle Gutierrez, Linda Otero, and Sean Scruggs. Official recognition of the CGTO comes from the 18 tribal governments who are formally consulting

with the U.S. Nellis Air Force Base, Nevada Test and Training Range in Southern Nevada. Funding for this and all other CGTO activities comes from the U.S. Air Force. Earlier studies that have been used for tiering in this analysis included members of the Department of Energy, Nevada Test Site, American Indian Transportation Committee: Richard Arnold, Pahrump Paiute Tribe, Elliot Booth, Colorado River Indian Tribes, Don Cloquet, Las Vegas Indian Center, Betty Cornelius, Colorado River Indian Tribes, Larry Eddy, Colorado River Indian Tribes, Maurice Frank, Yomba Shoshone Tribe, Milton Hooper, Confederated Tribes of the Goshute Reservation, Ted Howard, Shoshone-Paiute Tribe of the Duck Valley Reservation, Calvin Meyers, Moapa Paiute Indian Tribe and Gaylene Moose, Big Pine Paiute Tribe. Funds for their environmental assessment work were provided by the U.S. Department of Energy. Guidance and support for the LEIS ethnographic study was provided by Diane Austin, Chair of the School of Anthropology, University of Arizona. The research team included UofA students Christopher Sittler, Christopher Mintie Johnson, Mariah Albertie, Cameron Kays, and Grace Penry, and Daniel Velasco. Special thanks to Noah Pleshet, Visiting Associate in the SofA, who facilitated the last field session in December 2018.

**Conflicts of Interest:** The authors declare no conflict of interest.

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
