# Peer review of "Landscape Is Alive: Nuwuvi Pilgrimage and Power Places in Nevada"

_land, doi:10.3390/land11081208_

Round 1

Reviewer 1 Report

land-1746604-peer-review-v1

Landscape is Alive: Nuwuvi Pilgrimage and Power Places in Nevada

Overall this is a great paper that presents an in depth case study of cultural/spiritual special places as they manifest themselves  in networked set of trails. As such, then the paper substantially adds to the discussion of cultural landscapes in heritage management/historic preservation. The core of the pare is well written and certainly worth publishing. It can be predicted that it will be reasonable well cited.

While there is, understandably, a great deal of self-citation, especially by Stoffle, a critical engagement with the wider literature on cultural landscapes in heritage management/historic preservation is lacking.

As it stands, it is an in-depth ethnographic case study, which is fine in its on right, but not suited for LAND. For the audience of this journal, there needs to be a brief introductory section that strongly frames the paper in terms of cultural landscape research /theory and strongly signals the contributions this paper makes to the state of knowledge. The end section then needs to tie the case study back to the framing.

This is not difficult to do and will add to the value of the paper and its citeability.

Author Response

We have addressed the issues raised by the reviewer by explaining why there as been a great deal of self-citation. We have also included a discussion on the wider cultural landscape and heritage literature and how this essay ties into these issues to address the reviewer's concerns.

Reviewer 2 Report

The paper revolves around the concept of landscape. The main problem is that landscape as a re-theorized geographical concept is not discussed nor problematized at all. The paper is concerned mostly with an empirical discussion as may easily be seen in both the abstract the intro part and surely the conclusions. 

what is the merit of engaging in your vast and deep ethnographies if someone who is not from your area cannot understand the theoretical contribution of this paper save for local experts? 

Author Response

To address the reviewer's concerns, we have placed our discussion of this case into the wider discussion of cultural landscapes and heritage management. We have also explained in detail the research methods used in building our analysis.

Round 2

Reviewer 1 Report

In normal academic practice there should be a document that sets out point by point the responses to a reviewer's comments. In a submission with line numbers, that is easy to do. This did not occur.

On a substantive level, while some more historic level introductory comments have been made on cultural landscapes, the paper does not engage at all with the current theoretical literature on the topic. The short section added to the discussion is also inadequate and does not tie back to the the framing (as little as t is)m in the introduction. It is still not properly grounded and does not signal its relevance to the wider discourse. As noted in the initial review:

As it stands, it is an in-depth ethnographic case study, which is fine in its on right, but not suited for LAND. For the audience of this journal, there needs to be a brief introductory section that strongly frames the paper in terms of cultural landscape research /theory and strongly signals the contributions this paper makes to the state of knowledge. The end section then needs to tie the case study back to the framing. 

Also, the authors assert that "We have addressed the issues raised by the reviewer by explaining why there as been a great deal of self-citation. "  I must admit I fail to find reference to this in the revised paper. 

I recommend that the authors be invited to carry out the required major revision as recommended in the first round. I do so only as the core of the paper has values, else I would recommend reject at this stage.

The authors would be well advised to a reviewer's comments seriously.

Reviewer 2 Report

I have read the current corrected version and found it still very much

> wanting and lack of theoretical merit The authors are totally

> oblivious to theoretical notion of landscape in contemporary cultural

> geography As such I fail to see how this descriptive narrative

> broadens as they claim our understanding of landscape